# Nanoscale self-organization and metastable non-thermal metallicity in Mott insulators

Andrea Ronchi [1,2,3,8]✉, Paolo Franceschini[1,2,3,9], Andrea De Poli[1,3,4], Pía Homm[2], Ann Fitzpatrick [5], Francesco Maccherozzi[5], Gabriele Ferrini [1,3], Francesco Banfi [6], Sarnjeet S. Dhesi [5], Mariela Menghini [2,7], Michele Fabrizio [4]✉, Jean-Pierre Locquet[2] & Claudio Giannetti [1,3]✉

Mott transitions in real materials are first order and almost always associated with lattice distortions, both features promoting the emergence of nanotextured phases. This nanoscale self-organization creates spatially inhomogeneous regions, which can host and protect transient non-thermal electronic and lattice states triggered by light excitation. Here, we combine time-resolved X-ray microscopy with a Landau-Ginzburg functional approach for calculating the strain and electronic real-space configurations. We investigate $V_2O_3$, the archetypal Mott insulator in which nanoscale self-organization already exists in the low-temperature monoclinic phase and strongly affects the transition towards the high-temperature corundum metallic phase. Our joint experimental-theoretical approach uncovers a remarkable out-of-equilibrium phenomenon: the photo-induced stabilisation of the long sought monoclinic metal phase, which is absent at equilibrium and in homogeneous materials, but emerges as a metastable state solely when light excitation is combined with the underlying nanotexture of the monoclinic lattice.

[1] Department of Mathematics and Physics, Università Cattolica del Sacro Cuore, Brescia I-25133, Italy. [2] Department of Physics and Astronomy, KU Leuven, Celestijnenlaan 200D, 3001 Leuven, Belgium. [3] ILAMP (Interdisciplinary Laboratories for Advanced Materials Physics), Università Cattolica del Sacro Cuore, Brescia I-25133, Italy. [4] Scuola Internazionale Superiore di Studi Avanzati (SISSA), Via Bonomea 265, 34136 Trieste, Italy. [5] Diamond Light Source, Didcot, Oxfordshire OX11 0DE, UK. [6] FemtoNanoOptics group, Université de Lyon, CNRS, Université Claude Bernard Lyon 1, Institut Lumière Matière, F-69622 Villeurbanne, France. [7] IMDEA Nanociencia, Cantoblanco, 28049 Madrid, Spain. [8] Present address: Pirelli Tyre S.p.A, viale Piero e Alberto Pirelli 25, Milano 20126, Italy. [9] Present address: CNR-INO (National Institute of Optics), via Branze 45, 25123 Brescia, Italy. ✉email: andrea.ronchi@pirelli.com; fabrizio@sissa.it; claudio.giannetti@unicatt.it

Since its original proposal back in 1949[1], the Mott metal–insulator transition keeps attracting interest. For long, experimentalists and theorists have put a lot of effort to understand the microscopic bases of such transitions in real materials and model Hamiltonians. However, it has recently become urgent to extend that effort towards understanding the dynamics of the Mott transition on multiple timescales and at length scales much longer than the inter-atomic distances[2], which are most relevant for potential applications. Indeed, Mott transitions in real materials have a first-order character, often very pronounced, so that driving across such transitions requires one phase, formerly metastable, to nucleate, grow, and finally prevail over the other, formerly stable. In addition, real Mott transitions are nearly all the times accompanied by a lattice distortion, which, besides enhancing the first-order character of the transition, also constrains the nucleation and growth dynamics, fostering the emergence of nanotextures within the insulator-metal coexistence region[3,4]. Moreover, it may happen that also the lower symmetry crystal structure, usually the insulating phase, is inhomogeneous because of coexisting twins[5]. This is often the case when the elastic strain is directly involved in the structural transition.

Such circumstances might, at first sight, be regarded just as unwanted side effects that make the Mott transition more complex. In Mott materials, the onset of high-temperature metallicity is accompanied by the melting of the low-temperature lattice configuration, which is a slow process and implies a complex real-space rearrangement of domains at the nanoscale. This slow dynamics constitutes the bottleneck for the realization of electronic volatile switches operating at frequencies as high as several THz[6], which often discouraged potential applications of insulator-to-metal transitions (IMT) in real materials either driven by temperature changes or by non-equilibrium protocols such as light excitation. In this framework, specific efforts have been recently devoted to investigate possible transient non-thermal states in vanadium oxides, which undergo temperature-driven IMT of great interest for resistive switching and neuromorphic computing applications[6–21]. Much activity has focused on developing strategies to decouple the electronic and structural changes, with the ultimate goal of achieving all-electronic switching for ultrafast Mottronics. The recent claim of a photo-induced metallic phase of monoclinic $VO_2$[22,23] has triggered a huge effort to address to what extent the photo-induced transition is similar to the thermally driven one and whether the electronic and lattice degrees of freedom remain coupled at the nanoscale during and after the light excitation[3,20,24–26].

The goal of this work is to finally clarify the role of spatial nanotexture in controlling the Mott transition dynamics and in favouring the decoupling of the electronic and lattice transformations when the system is driven out-of-equilibrium by light pulses. We focus on the archetypal Mott insulator $V_2O_3$[27–30], which indeed realizes all at once the full complex phenomenology we previously outlined, and thus is the privileged playground to attempt such an effort. A nanotextured metal–insulator coexistence across the equilibrium first-order transition in thin films has in fact been observed by near-field infrared microscopy in ref. [3]. Specifically, the metal–insulator coexistence is characterized by a rather regular array of striped metallic and insulating domains oriented along two of the three possible hexagonal axes of the high-temperature rhombohedral structure, the missing twin possibly being a consequence of the R-plane orientation of the film[31]. Later, it has been observed, still on thin films but now with a c-plane orientation and using X-ray Photoemission Electron Microscopy (PEEM)[32], that the monoclinic insulator is itself nanotextured. In particular, such a phase looks like a patchwork of the three equivalent monoclinic twins oriented along the three hexagonal axes of the parent rhombohedral phase. Upon raising the temperature of the monoclinic insulator, metallic domains start nucleating along the interfaces between the monoclinic twins[32], thus forming stripes coexisting with insulating ones, all of them again oriented along the hexagonal axes, in agreement with the experiment in ref. [3]. The origin of this complex nanoscale self-organization, which was tentatively attributed to the long-range Coulomb repulsion[3], still remains unexplained.

In this work, we develop a coarse-grained approach that is able to capture the real-space lattice and electron dynamics of the IMT in $V_2O_3$. Our model demonstrates that the intrinsic nanotexture is driven by the elastic strain associated with the monoclinic lattice distortion. The full understanding of the transition dynamics also discloses the possibility of stabilizing a non-thermal metallic electronic state, which retains the insulating monoclinic lattice structure. This state is unfavourable at equilibrium and in homogeneous materials, but it can be photo-induced when the electronic population within the vanadium 3d bands is modified by ultrafast light pulses. The intrinsic nanotexture is key to create the strain conditions at the boundaries of the monoclinic twins, which protect and stabilize the non-thermal monoclinic metallic phase. We experimentally demonstrate the existence of such a metastable phase by performing novel synchrotron-based time-resolved X-ray PEEM (tr-PEEM) experiments with 30 nm and 80 ps spatial and temporal resolution. The excitation of $V_2O_3$ thin films with intense infrared (1.5 eV) ultrashort light pulses turns the material into a metal with the same shear-strain nanotexture of the insulating phase. Even though all experimental and theoretical results we are going to present refer to vanadium sesquioxide, they reveal an unexpected richness that may as well emerge in other Mott insulating materials. The role played by the spontaneous nanoscale lattice architectures characterizing first-order IMT provides a new parameter to achieve full control of the electronic phase transformation in Mott materials.

The work is organized as follows. Firstly, we provide an overview of the lattice and electronic transformations which characterize the phase diagram of $V_2O_3$, as well as a characterization of the spontaneous nanotexture of the monoclinic insulating phase. This information is crucial since it provides the microscopic bases of the multiscale model of the lattice and electronic transition. Secondly, we introduce the model, based on proper Landau–Ginzburg functionals, and we show how it captures the nanotexture formation as well as the dynamics of the temperature-induced phase transition. Finally, we present the central non-equilibrium results. The multiscale model is extended to treat the non-equilibrium case. The model shows that the nanotexture can favour and stabilize a non-thermal electronic metallic phase which retains the monoclinic shear strain of the low-temperature lattice. We experimentally demonstrate this phenomenon by performing tr-PEEM measurements on $V_2O_3$ thin films disclosing the ultrafast dynamics with spatial resolution. Finally, we also show the possibility of controlling the non-thermal transition dynamics by interface strain engineering.

**The $V_2O_3$ phase diagram and lattice transformation.** The phase diagram of $(V_{1-x}M_x)_2O_3$, $M = $ Cr,Ti, is shown in Fig. 1a. It includes rhombohedral paramagnetic insulator and metal phases with corundum structure, and a low-temperature dome where the system is a monoclinic antiferromagnetic Mott insulator. The effect of pressure is, as expected, to favour the metal phase, alike that of Ti doping, although, above 32.5 GPa, such metal appears to be also monoclinic at room temperature[33,34]. The low-pressure transition from the high-temperature corundum structure to the low-temperature monoclinic one has a first-order nature that weakens with Cr doping x, and maybe turns continuous above $x \simeq 0.03$. For pure $V_2O_3$, the case of interest here, dashed vertical line in Fig. 1a, the electronic, magnetic and structural transition

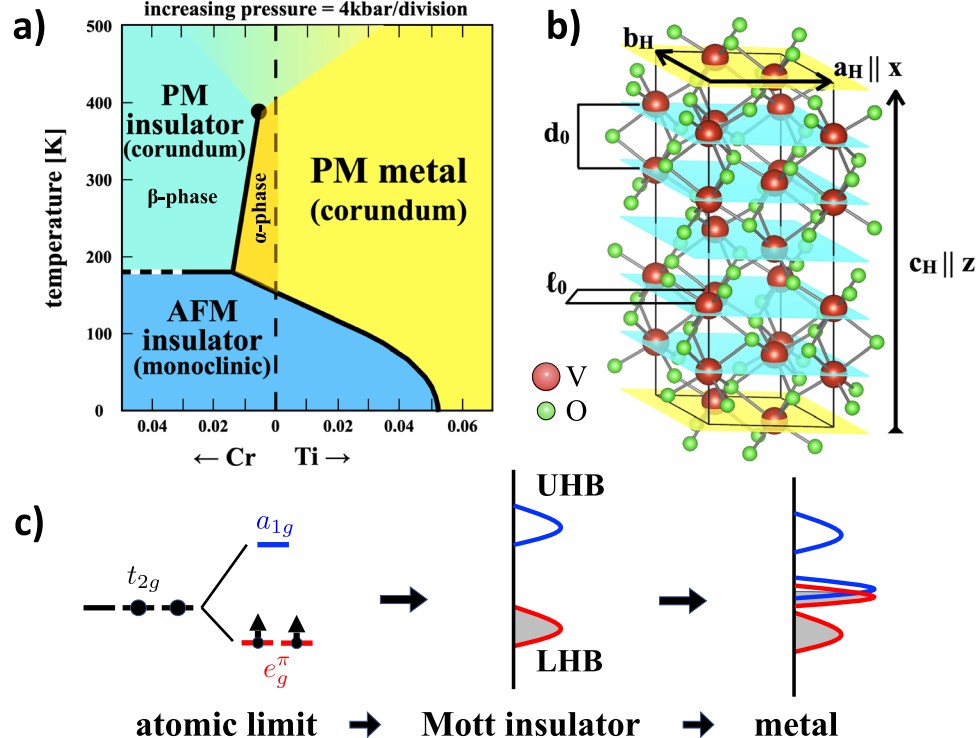

**Fig. 1 Phase diagram and corundum structure. a** Phase diagram of $(V_{1-x}M_x)_2O_3$, with $M =$ Cr, Ti, as a function of the doping concentration $x$ and pressure, from ref. [29]. AFM and PM stand for antiferromagnetic and paramagnetic, respectively. All transition lines (solid black lines) are first order, with the one separating PM metal from PM insulator that terminates into a second-order critical point (black dot). The paramagnetic metal and insulating phases of Cr-doped $V_2O_3$ are commonly referred to as $\alpha$ and $\beta$ phases, respectively, and shown in the figure. **b** Non-primitive hexagonal unit cell of the high-temperature corundum phase. The V–V nearest neighbour distance within the $a_H$–$b_H$ plane, $\ell_O$, and along $c_H$, $d_O$, are also shown. Also shown is the reference frame we use throughout the work, with $\mathbf{a}_H\|\mathbf{x}$ and $\mathbf{c}_H\|\mathbf{z}$. **c** Sketch of the electronic Mott transition in $V_2O_3$. In the atomic limit, the two conduction electrons of $V^{3+}$ occupy the $t_{2g}$ orbital of the cubic-field split 3d-shell. Because of the additional trigonal distortion, the $t_{2g}$ is further split into a lower $e_g^\pi$ doublet and higher $a_{1g}$ singlet. The two electrons thus sit into the $e_g^\pi$ orbital, in a spin triplet configuration because of Hund's rules. In the solid the atomic levels corresponding to removing or adding one electron broaden into lower and upper Hubbard bands, LHB and UHB, respectively. The LHB has prevailing $e_g^\pi$ character while the UHB has dominant $a_{1g}$ character[41]. Note that we do not show, for simplicity, the multiplet structure that the Hubbard bands must have because of Coulomb exchange splitting. In the metal phase, overlapping quasiparticle bands appear at the Fermi level.

that occurs at $T_c \simeq 170$K in bulk crystals has a very pronounced first-order character: the jump in resistivity covers almost six orders of magnitude[28], and the strain-driven rhombohedral-monoclinic martensitic transformation[35] can be rather destructive if the sample is not dealt with care.

In the rhombohedral phase above $T_c$, $V_2O_3$ crystallizes in a corundum structure, space group $R\bar{3}c$ No. 167. The non-primitive hexagonal unit cell contains six formula units, and has the lattice vectors shown in Fig. 1b, where[28,36,37]

$$a_H = b_H \simeq 4.936\,\text{Å}, c_H \simeq 14.021\,\text{Å}, \quad (1)$$

and, by convention, we choose $\mathbf{a}_H\|\mathbf{x}$ and $\mathbf{c}_H\|\mathbf{z}$. The vanadium atoms form honeycomb planes with ABC stacking, see also Supplementary Note 2 (See the Supplementary Material). It follows that each Vanadium has only one nearest neighbour along the hexagonal $c_H$-axis. We hereafter denote such vertical pairs as dimers. Moreover, the two inequivalent V atoms within each honeycomb plane do not lie on such plane, see Fig. 1b: the atoms that form dimers with the plane above/below are shifted down/up. The dimer bond length $d_0$ is slightly shorter than the distance $\ell_0$ between nearest neighbour V atoms within the hexagonal planes, specifically

$$d_0 \simeq 2.7\,\text{Å}, \ell_0 \simeq 2.873\,\text{Å}. \quad (2)$$

Such difference reflects a trigonal distortion of the oxygen octahedra surrounding each vanadium, which is responsible for

the V-3d $t_{2g}$ orbital splitting into a lower $e_g^\pi$ doublet and an upper $a_{1g}$ singlet, and is believed to play a crucial role in the Mott metal–insulator transition[38]. In Fig. 1c we draw an oversimplified picture of the Mott transition to emphasize the role of the trigonal crystal field splitting.

The magnetic insulator below $T_c$ has a monoclinic crystal structure, space group $I2/a$, No. 15. The structural distortion breaks the $C_3$ rotation symmetry around the $\mathbf{c}_H$-axis, and can be viewed[36] as a rotation of the atoms in a plane perpendicular to one of three hexagonal axes, $\mathbf{a}_H$, $\mathbf{b}_H$ and $-\mathbf{a}_H-\mathbf{b}_H$. The three choices correspond to equivalent monoclinic structures, which are distinguishable only in reference to the parent corundum state. We shall here choose for simplicity the $\mathbf{a}_H\|\mathbf{x}$ rotation axis, which becomes the monoclinic primitive lattice vector $\mathbf{b}_m$, so that the rotation occurs in the $\mathbf{y}$–$\mathbf{z}$ plane, where the monoclinic lattice vectors $\mathbf{a}_m$ and $\mathbf{c}_m$ lie. Concerning magnetism, each $\mathbf{a}_m$–$\mathbf{c}_m$ plane is ferromagnetic, while adjacent planes are coupled to each other antiferromagnetically, see Fig. 2a, b. In other words, the dimers, which do lie in the $\mathbf{a}_m$–$\mathbf{c}_m$ plane, are ferromagnetic. Similarly, of the three nearest neighbour bonds in the hexagonal plane, the one lying in the $\mathbf{a}_m$–$\mathbf{c}_m$ plane of length $\ell_y$ is therefore ferromagnetic, while the other two, of length $\ell_1$ and $\ell_2$, are antiferromagnetic, see Fig. 2a.

Given our choice of the monoclinic twinning, the relation between monoclinic and hexagonal lattice vectors[28,36] are graphically shown in Fig. 2a, b. However, for later convenience,

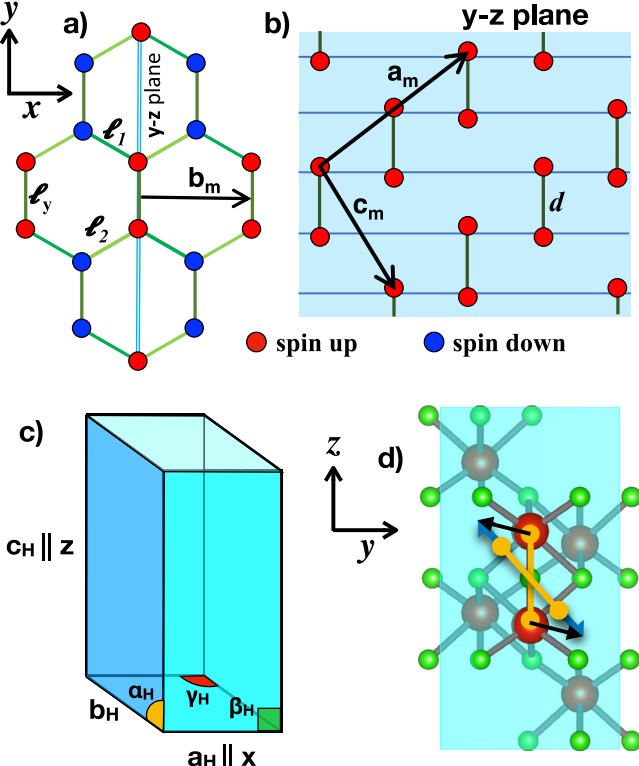

**Fig. 2 Magnetic and structural properties of the monoclinic phase.**
Magnetic order in the hexagonal plane (**a**) and in the **y**–**z** plane, equivalent
to the $\mathbf{a}_m$–$\mathbf{c}_m$ one (**b**). In the hexagonal plane we also draw the nearest
neighbour bond lengths: $l_y$, which is ferromagnetic, and $l_1$ and $l_2$, both
antiferromagnetic. The dimers, with bond length $d$, lie in the **y**–**z** plane and
are ferromagnetic, see (**b**), where we also show the monoclinic lattice
vectors $\mathbf{a}_m$ and $\mathbf{c}_m$. **c** The pseudo-hexagonal unit cell that we use throughout
this work. **d** The dimer, in yellow, which in the rhombohedral phase lies
along $\mathbf{z}\|\mathbf{c}_H$, rotates anticlockwise around $\mathbf{x}\|\mathbf{b}_m = \mathbf{a}_H$ and elongates, so that
the vanadium atoms at the endpoints move towards the octahedral voids.

we hereafter prefer to use a non-primitive pseudo-hexagonal
unit cell, see Fig. 2c. With such choice the monoclinic phase
corresponds to

$$a_H \simeq 5.002\,\text{Å}, \quad b_H \simeq 4.974\,\text{Å}, \quad c_H \simeq 13.953\,\text{Å},$$
$$\alpha_H \simeq 91.73°, \qquad \beta_H = 90°, \qquad \gamma_H \simeq 120.18°, \tag{3}$$

as opposed to the corundum parameters in Eq. (1), with
$\alpha_H = \beta_H = 90°$ and $\gamma_H = 120°$. The corundum-to-monoclinic
transition is therefore accompanied by a volume expansion of
1.4%[28], as expected across a metal–insulator transition.

Considering the nearest neighbour V–V distances in the
hexagonal plane, $l_y$, $l_1$, and $l_2$, and the dimer length $d$, see Fig. 2,
in the monoclinic phase as compared to the rhombohedral one
($d_0$), see Eq. (2), $l_y$ grows by 4%, $d$ by 1.6%, while $l_1$ and $l_2$ are
almost unchanged, one is 0.38% shorter and the other 0.14%
longer. In other words, in accordance with the Goodenough-
Kanamori-Anderson rules, all ferromagnetic bonds lengthen, the
planar one $l_y$ quite a bit more than the dimer.

## Results and discussion
### Spontaneous nanotexture of the monoclinic insulating phase.
As described in ref. [32], equilibrium photo-emission electron
microscopy (PEEM), exploiting X-ray linear dichroism (XLD)[39]
as the contrast mechanism, can be used to investigate the
dynamics of the rhombohedral to monoclinic transition in real
space. As shown schematically in Fig. 3a, Linear Horizontal (LH)

*s*-polarized X-ray pulses in resonance with the V $L_{2,3}$-edge (520
eV) impinge with 75° incident angle on a 40 nm $V_2O_3$ film epi-
taxially grown on a (0001)-$Al_2O_3$ substrate, therefore with the $c_H$-
axis oriented perpendicular to the surface of the film[40]. In order
to optimize the signal and remove possible backgrounds and
artifacts, each reported image is the difference between images
taken with X-ray pulses at 520 eV, for which the contrast between
signals from different possible monoclinic distortions is max-
imum, and 518 eV, for which the contrast is minimum[32,41]. In
Fig. 3b we report typical spatially resolved XLD-PEEM images
taken at $T = 100$ K, i.e., fully in the monoclinic insulating phase
of $V_2O_3$. For each direction of the X-rays polarization (see the
grey arrows on top of the images) three different striped insu-
lating nano-domains, indicated with red, blue and yellow colours,
are clearly visible. The stripe-like domains evidently follow the
hexagonal symmetry of the undistorted rhombohedral phase,
with characteristic dimensions of a few micrometres in length and
200–300 nm in width. The observed spontaneous nanotexture of
the monoclinic phase demonstrates that the minimization of the
total elastic energy drives the formation of domains in which the
monoclinic distortion takes place along one of the three equiva-
lent directions. As extensively discussed in ref. [32], when the sys-
tem is heated up and the coexistence region is entered,
rhombohedral metallic droplets start nucleating at the domain
boundaries. Upon further heating up, the domains grow until the
insulator-to-metal percolative transition takes place at a metallic
filling fraction of the order of 0.45. We mentioned that each
monoclinic domain is uniquely identified by the monoclinic $\mathbf{b}_m$,
which can be any of the primitive hexagonal vectors, $\mathbf{a}_H$, $\mathbf{b}_H$ or
$-\mathbf{a}_H-\mathbf{b}_H$, and thus define three axes at 60° from each other. That
is revealed by rotating the X-ray polarization and plotting the
XLD signal integrated over specific regions (pink square, green
circle and yellow triangle in Fig. 3b corresponding to the three
different domains. As shown in Fig. 3c, the XLD signal of the
three different domains is indeed phase-shifted by 60° and has a
180° periodicity.

To interpret these observations, in Supplementary Note 3 (See
the Supplementary Material) we calculate the polarization
dependence of the XLD contrast within the monoclinic phase.
In brief, the $a_{1g}$ singlet and the $e_g^\pi$ doublet transform, respectively,
as the one dimensional, $A_1 \sim z^2$, and two dimensional, $E \sim (x,y)$,
irreducible representations of $D_3$. The monoclinic distortion
generates a mixing between $a_{1g}$ and the combination of the $e_g^\pi$
that lies on the $\mathbf{a}_m$–$\mathbf{c}_m$ monoclinic plane, i.e., perpendicular to $\mathbf{b}_m$.
Such combination evidently changes among the three equivalent
monoclinic twins, and, because of its directionality, it does
contribute to the XLD signal, which we predict is minimum
(maximum) for in-plane components of the field parallel
(perpendicular) to the $\mathbf{b}_m$ axis. Since the latter can be any of
the three hexagonal axes, that immediately explains the observa-
tion in Fig. 3 and allows inferring the direction of primitive
hexagonal vectors, which are shown in the figure. In addition, the
data reported in Fig. 3 make us conclude that the interface
between two monoclinic twins forms along the direction
perpendicular to the $\mathbf{b}_m$ axis of the third one.

### Multiscale modelling.
The self-organization of the monoclinic
insulating phase on length scales of the order of hundreds
nanometres calls for a multiscale theory effort which builds on
the microscopic parameters governing the transition but goes
beyond by describing the formation of domains at length scales
much larger than the lattice unit cell. Already Denier and
Marezio[36] emphasized that the main change that occurs across
the rhombohedral to monoclinic transition is actually the dis-
placement of the V atoms of a dimer towards "the adjacent

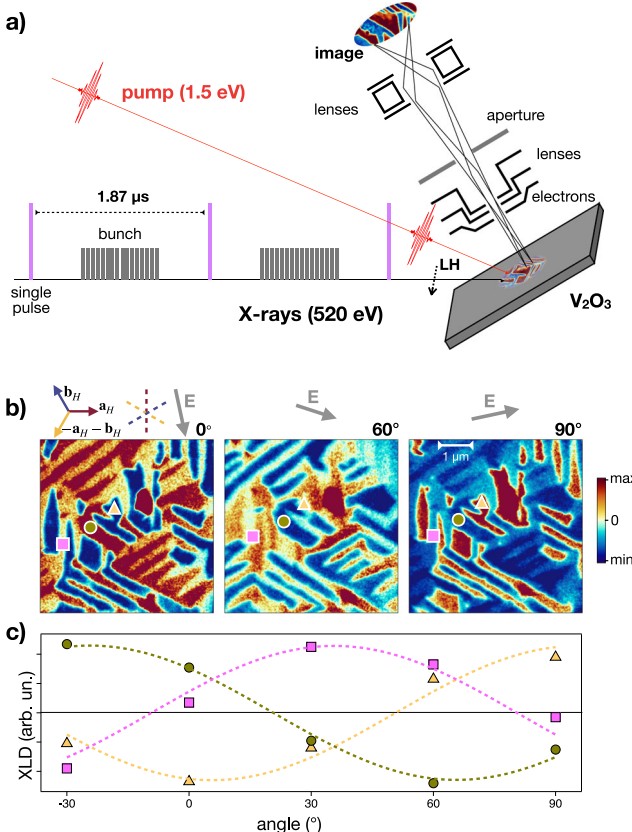

**Fig. 3 X-ray microscopy. a** Experimental setup. X-rays bunches with tunable energy resonant with the vanadium $L_{2,3}$ edge impinge on the sample at ~75° incidence. The X-ray linear polarization can be rotated from in-plane (Horizontal) to out-of-plane (Vertical). The electrons emitted from the sample are collected and imaged through electrostatic lenses. In the time-resolved configuration, the signal originated by isolated X-rays pulses with linear horizontal (LH) polarization is collected by suitable synchronized gating of the detection apparatus. The pump infrared laser is synchronized to the synchrotron pulses. **b** XLD-PEEM images taken at 100 K evidencing striped monoclinic domains characterized by different XLD intensity. The three panels show images taken for different polarization angles of the impinging X-ray pulses. The indicated angle refers to the initial (left panel) polarization, which is taken as reference. The colour scale indicates the amplitude (arb. units) of the PEEM signal. We note the existence of the three distinct domains (red, blue, yellow), namely the three monoclinic twins. The XLD signal, as demonstrated in Supplementary Note 3 (See the Supplementary Material), is minimum (blue scale in the panels) when the electric field (grey arrows on top of the images) is parallel to the monoclinic $\mathbf{b}_m$ axis, which can be any of the primitive hexagonal vectors, $\mathbf{a}_H$, $\mathbf{b}_H$ or $-\mathbf{a}_H-\mathbf{b}_H$, and maximum when $\mathbf{E} \cdot \mathbf{b}_m = 0$. Comparing the theoretical prediction with the data, we infer the three primitive hexagonal vectors shown in the figure, and further conclude that that the interface between two twins is perpendicular to the $\mathbf{b}_m$ axis of the third domain. **c** XLD signal as a function of the angle between the X-ray polarization and the sample axis. The pink square, green circle and yellow triangle refer to the positions indicated in (**a**). Within each domain, the XLD signal displays the expected 180° periodicity. When comparing the three distinct domains, the XLD signal shows the predicted 60° phase shift.

octahedral voids", see Fig. 2d and also Supplementary Note 2 (See Supplementary Material). Such displacement results in a $\theta = 1.8°$ anticlockwise rotation of the dimers around the $\mathbf{b}_m$ axis, thus the value of $\alpha_H \simeq 91.73°$ and the $c_H$ axis compression in Eq. (3). In addition, the dimer elongates by 1.6%. More specifically, the

Vanadium displacement, with our choice of monoclinic axis $\mathbf{b}_m \| \mathbf{x}$, has non-negligible components along both $\mathbf{y}$ and $\mathbf{z}$, the former leading to the 1.8° dimer tilting, and the latter mostly responsible for the 1.6% stretching of the bond. We note that the tilting alone accounts not only for the variations of $\alpha_H$ and $c_H$, but also for the increase in $\ell_y$, which is the most significant change crossing the structural transition, as well as for the dilation along the hexagonal $a_H$, cf. Eq. (1) with (3). Therefore, the deformation of the unit cell across the transition is primarily a consequence of the dimer tilting, namely of the Vanadium displacements along $\mathbf{y}$.

Hereafter, we thus make the assumption that the $\mathbf{y}$ and $\mathbf{z}$ components of the vanadium displacement, or, equivalently, the dimer tilting and its elongation, correspond to different degrees of freedom, by all means coupled to each other[42], but each playing its distinctive role.

The tilting is ultimately responsible for breaking the threefold rotation symmetry around $\mathbf{c}_H$, and thus the rhombohedral to monoclinic transition, which may not necessarily go along with a metal–insulator transition. DFT-GGA electronic structure calculations, which are supposedly valid at weak-coupling, predict[43] that the Fermi surface of the corundum metal is unstable towards a monoclinic distortion. However, that same instability is also found in the opposite strong-coupling limit. Indeed, assuming a Mott insulating state in which each Vanadium acts like a spin-1, see Fig. 1c, it was shown[43] that $V_2O_3$ realizes on each honeycomb plane a highly frustrated Heisenberg model, with comparable nearest and next-nearest neighbour antiferromagnetic exchange constants[44–46]. Such frustration is efficiently resolved by the monoclinic distortion stabilizing the stripe phase shown in Fig. 2a. This prediction has got further support from recent inelastic neutron scattering data combined with DFT calculations[47]. All the above results suggest that the corundum phase of pure $V_2O_3$ is intrinsically unstable, and destined to turn into a monoclinic phase at low temperature. The fact that such transition happens to coincide with a metal to an antiferromagnetic insulator transition indicates just a strong positive interplay between the lattice instability and the electronic correlations[42,43,47]. However, nothing would prevent the structural and the metal–insulator transitions to occur separately. Indeed, chromium doping, see Fig. 1, does drive a metal–insulator transition without an intervening monoclinic distortion. The opposite case of a monoclinic transition not accompanied by a metal–insulator one is still highly debated. Evidences of a monoclinic metal phase have been observed in $V_2O_3$ at high pressure[33,34]. In contrast, at ambient pressure the possible existence of a monoclinic metal phase remains so far controversial[3,26,32,48–53].

The vanadium displacement along $\mathbf{z}$ preserves the $R\bar{3}c$ space group but lengthens the dimers, thus weakening their bonding strength and increasing the trigonal field splitting between lower $e_g^\pi$ and upper $a_{1g}$. Both those effects are believed[38] to drive $V_2O_3$ towards a Mott insulating state irrespective of the monoclinic distortion, even though how they precisely affect the low-energy single-particle spectrum approaching the transition is still under debate[38,54]. However, the existence of a strong connection between the $\mathbf{z}$-displacement of vanadium and the Mott transition is undeniable. That connection can be more neatly inferred comparing, see Fig. 1, the rhombohedral paramagnetic Cr-doped insulator, so-called $\beta$-phase, with the rhombohedral paramagnetic metal phase of pure $V_2O_3$, or of the weakly Cr-doped metallic $\alpha$-phase.

Indeed, the most noticeable structural change that occurs at room temperature in the insulating $\beta$-phase as opposed to the metallic phases is the ~15−16%[55,56] of the parameter $z$ that identifies the Wyckoff position 12c of V in the $R\bar{3}c$ space group. Physically, $z$ determines the dimer length $d_0 = c_H(2z-1/2)$, see Supplementary Note 2 (See the Supplementary Material),

which significantly increases in the $\beta$-phase[55,56]. This observation suggests that $z$, or, equivalently, the dimer length, must be intimately correlated to the metal versus insulator character of Cr-doped with respect to pure $V_2O_3$, otherwise it is not comprehensible why $d_0$ increases despite $c_H$ decreases[55,56]. Moreover, larger values of $d_0$ presumably correspond to larger orbital polarization $\Delta n = n_{e_g^\pi} - n_{a_{1g}}$[57], consistent with the evidence that $\Delta n$ grows across the metal-to-insulator transition[41,58]. The only exception is the pressure-driven transition from the $\beta$-insulator to the $\alpha$-metal, across which $\Delta n$ does not show appreciable changes[58]. In that case, however, the behaviour of the dimer length across the transition is not known, and the homogeneity of the $\alpha$-metal not guaranteed[59,60].

We shall here give up any attempt to describe microscopically the rhombohedral-to-monoclinic and metal-to-insulator transitions in $V_2O_3$, and instead resort to a more macroscopic approach based on a Landau–Ginzburg theory for the order parameters that characterize, respectively, the two transitions. As we discussed, the order parameter associated with the metal–insulator transition can be legitimately identified with the dimer length $d$. We emphasize that such identification by no means implies that $d$ alone drives the transition, which is mainly of electronic origin, but just that the behaviour of $d$ mimics faithfully that of resistivity, and thus, in the spirit of a Landau–Ginzburg approach, that $d$ can be rightfully promoted to the order parameter of that transition. The tilting of the dimer is instead directly linked to the shear elements $\epsilon_{13}$ and $\epsilon_{23}$ of the strain tensor, which behave like the components of a planar vector

$$\boldsymbol{\epsilon}_2 = \left(\epsilon_{13}, \epsilon_{23}\right) \equiv \epsilon_2 \left(\cos\phi_2, \sin\phi_2\right), \quad (4)$$

see Supplementary Note 2 (See the Supplementary Material). We can therefore associate $\boldsymbol{\epsilon}_2$ with the two-component order parameter of the rhombohedral to monoclinic transition. Specifically, using the structural data of the corundum and monoclinic structures across the transition[28,36,37], and in our reference frame, the shear-strain order parameter $\boldsymbol{\epsilon}_2$ just after the transition has magnitude $\epsilon_2 \simeq 0.01756$, and three possible orientations, i.e., monoclinic twins, defined by the phases, see Eq. (4),

$$\phi_{2,n} = \frac{\pi}{6} + (n-1)\frac{2\pi}{3}, \quad n = 1, 2, 3. \quad (5)$$

In other words, $\boldsymbol{\epsilon}_2$ is directed along $\mathbf{b}_m \wedge \mathbf{c}_H$.

The Landau–Ginzburg functional that we are going to present and study is therefore just the Born–Oppenheimer effective potential of the shear strain $\boldsymbol{\epsilon}_2$ and dimer length $d$. As such, that potential is substantially contributed by the electrons, though the latter does not appear explicitly. As earlier mentioned, we do not attempt to derive the Born–Oppenheimer potential by directly tracing out the microscopic electronic degrees of freedom[61–63]. In contrast, we build the Landau–Ginzburg functional by rather general arguments, mainly based on symmetry as well as on the phase diagram, Fig. 1, and the physical properties of $V_2O_3$. In addition, we infer the behaviour, in temperature and Cr-doping, of all parameters that characterize the Born–Oppenheimer potential through experiments, which are fortunately abundant for this compound.

**Landau–Ginzburg theory of the structural transition**. A Landau–Ginzburg free-energy functional for the space-dependent order parameter $\boldsymbol{\epsilon}_2(\mathbf{r})$ is rather cumbersome to derive, since all other components of the strain, beside the order parameter, are involved, and need to be integrated out to obtain a functional of $\boldsymbol{\epsilon}_2(\mathbf{r})$ only. This is further complicated by the constraints due to the Saint-Venant compatibility equations, which avoid gaps and

overlaps of different strained regions and play a crucial role in stabilizing domains below the martensitic transformation[64–66]. Therefore, not to weigh down the text, we present the detailed derivation of the Landau–Ginzburg functional in Supplementary Note 4 (See the Supplementary Material), and here just discuss the final result in the $c$-plane oriented film geometry of the experiments.

We find that the shear-strain two-component order parameter $\boldsymbol{\epsilon}_2(\mathbf{r})$ is controlled by the energy functional (See the Supplementary Material)

$$E[\boldsymbol{\epsilon}_2] = \int d\mathbf{r}\left\{-\frac{K}{2}\boldsymbol{\epsilon}_2(\mathbf{r})\cdot\nabla^2\boldsymbol{\epsilon}_2(\mathbf{r}) + \tau\,\epsilon_2(\mathbf{r})^2 - \gamma\,\epsilon_2(\mathbf{r})^3\,\sin 3\phi_2(\mathbf{r}) + \mu\,\epsilon_2(\mathbf{r})^4\right\}$$
$$+ \kappa \iint d\mathbf{r}\,d\mathbf{r}'\,\boldsymbol{\epsilon}_2(\mathbf{r})\cdot\hat{U}(\mathbf{r}-\mathbf{r}')\,\boldsymbol{\epsilon}_2(\mathbf{r}'), \quad (6)$$

where $\mathbf{r} = r\left(\cos\phi, \sin\phi\right)$ is the two-dimensional coordinate of the film, $K > 0$ is the strain stiffness, $\mu > 0$ the strength of the standard anharmonic quartic term that makes $E$ bounded from below, $\kappa$ a positive parameter that depends on the elastic constants $c_{11}$ and $c_{22}$ (See the Supplementary Material), while $\gamma$ is directly proportional to the elastic constant $c_{14}$ (See the Supplementary Material). The quadratic coupling constant $\tau \sim c_{44}$ encodes the electronic effects that, as earlier mentioned, make the corundum structure unstable at low temperature towards a monoclinic distortion, and thus drives the transition. Therefore, $\tau$ is positive in the high-temperature corundum phase and negative in the low-temperature monoclinic one. We emphasize that the above elastic constants refer to the $R\bar{3}c$ space group, so that, e.g., $\tau \sim c_{44} < 0$ simply means that the rhombohedral phase has become unstable. The last term in (6) derives from the Saint-Venant compatibility equations (See the Supplementary Material). Specifically,

$$\hat{U}(\mathbf{r}) = \begin{pmatrix} U_{11}(\mathbf{r}) & U_{12}(\mathbf{r}) \\ U_{21}(\mathbf{r}) & U_{22}(\mathbf{r}) \end{pmatrix}, \quad (7)$$

where the long-range kernels have the explicit expressions:

$$U_{11}(\mathbf{r}) = -U_{22}(\mathbf{r}) = -\frac{\cos 4\phi}{\pi\,r^2},$$
$$U_{12}(\mathbf{r}) = U_{21}(\mathbf{r}) = \frac{\sin 4\phi}{\pi\,r^2}, \quad (8)$$

and favour the existence of domains in the distorted structure[64–66]. The orientation of the interfaces between those domains is instead determined by the additional constraint we must impose (See the Supplementary Material) to fulfil the Saint-Venant equations, namely the curl-free condition

$$\nabla \wedge \boldsymbol{\epsilon}_2(\mathbf{r}) = 0. \quad (9)$$

If $\tau < 0$, the energy (6) is minimum at the angles defined in Eq. (5) when $\epsilon_2 > 0$. The constraint (9) implies that a sharp interface between two domains, identified by two of the three possible directions of $\boldsymbol{\epsilon}_2$, is oriented along the third one, namely along a mirror plane of the space group $R\bar{3}c$. This is exactly what is found experimentally, as we earlier discussed.

We also note that the minima of the energy functional (6) depend on the values and signs of $\tau$ and $\gamma \propto c_{14}$. Physically, $c_{14} > 0$ implies that an anticlockwise/clockwise rotation of the dimers in the $\mathbf{a}_m$–$\mathbf{c}_m$ plane drives an expansion/compression of the $\mathbf{b}_m$ axis, and vice versa for $c_{14} < 0$.

The corundum phase, $\boldsymbol{\epsilon}_2 = 0$, is a local minimum for $\tau > \tau_r \gtrsim 0$, $\tau_r$ being the rhombohedral spinodal point. Similarly, a monoclinic phase is a local minimum for $\tau < \tau_m$, where the monoclinic spinodal point $\tau_m \geq \tau_r$. Phase coexistence thus occurs when $\tau \in [\tau_r, \tau_m]$, and also the structural transition must take place at $\tau_c$ within that same interval. At ambient temperature and pressure, $\tau \sim c_{44} \simeq 53$ eV/cm$^3$, which must be greater than $\tau_c$ since the stable phase is corundum, and $c_{14} \simeq -12.5$ eV/cm$^3$ [67–69], so that $\gamma$ is negative, too. Well below

the structural transition at ambient pressure, $\tau$ must be smaller than $\tau_r$. Moreover, since the observed monoclinic distortion corresponds to an anticlockwise rotation of the dimers and an expansion of $\mathbf{b}_m$, the low-temperature value of $\gamma$ has to be positive. Therefore, upon lowering temperature $T$, $\tau$ must decrease, while $c_{14} \propto \gamma$ must increase and cross zero. There are actually evidence, specifically in chromium doped compounds[68], that $c_{14}$ does change sign approaching the transition from the corundum phase. We are not aware of any experimental measurement of $c_{14}$ in pure $V_2O_3$ below room temperature. Therefore, here we can only conjecture what may happen. One possibility is that $c_{14}$ crosses zero right at the structural transition, $\tau = \tau_c$. In such circumstances, the transition may become continuous[70], which is likely the case of Cr doping with $x \gtrsim 0.03$[71]. The alternative compatible with the ambient pressure phase diagram in Fig. 1 for pure or weakly Ti/Cr doped $V_2O_3$ is that $c_{14}$ becomes positive at temperatures higher than $T_c$, thus the observed first-order transition from the corundum phase, $\epsilon_2 = 0$, to the monoclinic one, $\epsilon_2 > 0$. There is still a third possibility that $\tau$ crosses $\tau_c$ at high temperature when $c_{14} \propto \gamma$ is still negative. In that case, the monoclinic phase that establishes for $\tau < \tau_c$ corresponds to a shear strain different from that observed in $V_2O_3$ at ambient pressure, specifically to a dimer tilting in the clockwise direction or, equivalently, to opposite $\boldsymbol{\epsilon}_2$ vectors. This is presumably what happens in the monoclinic metal phase observed above 32.5 GPa at 300 K[33].

Here, we shall not consider such extreme conditions, and therefore take for granted that, at ambient pressure and for high temperatures such that $\gamma$ is negative, $\tau$ remains greater than $\tau_c$, so that the stable phase is always rhombohedral. Moreover, since we are just interested in pure $V_2O_3$, we shall assume that $\gamma$ becomes positive well above the structural transition, around which we shall therefore consider $\gamma \sim c_{14} > 0$ constant, and $\tau \propto (T - T_0)$, with $T_0 > 0$ a parameter playing the role of a reduced temperature.

**Contribution from the dimer stretching**. The energy functional (6) controls only the shear strain $\boldsymbol{\epsilon}_2$, namely the dimer tilting. We still need to include the contribution of the dimer length $d$, which is assumed to play the role of the metal–insulator transition order parameter. For that, it is more convenient to start with the iso-structural paramagnetic metal and paramagnetic insulator phases, $\alpha$ and $\beta$, respectively, of Cr-doped $V_2O_3$, and only after include the coupling to the shear-strain order parameter $\boldsymbol{\epsilon}_2$. We note from Fig. 1 that the transition from the $\alpha$ phase to the $\beta$ one is first order, across which $d$ suddenly rises, occurs at a temperature $T_{\text{IMT}}(x)$ that depends on the Cr-doping $x$, and terminates at a second order critical point; a phenomenology reminiscent of a liquid-gas transition, as predicted[72,73] and experimentally confirmed[74]. In view of that analogy, we can reasonably assume that $d$ feels a double-well potential, the well centred at the smaller $d_M$ representing the metal and the other at $d_I > d_M$ the insulator, in presence of a symmetry breaking term that lowers either of the wells and depends on $T$ and $x$.

When the dimers rotate, the vanadiums at the endpoints tend to move towards the octahedral voids, and thus $d$ increases, see Fig. 2d. This suggests that a finite shear strain $\epsilon_2$ acts as a further symmetry breaking term, which lowers the potential well at larger $d$, in this way favouring the insulating phase. We mention that the positive interplay between the dimer tilting and its lengthening has been convincingly demonstrated by Tanaka in ref. [42].

Therefore, if we replace $d$ with the dimensionless space-dependent field

$$\eta(\mathbf{r}) = \frac{1}{d_I - d_M}\left(d(\mathbf{r}) - \frac{d_I + d_M}{2}\right), \tag{10}$$

so that $\eta = -1/2$ when $d = d_M$ and $\eta = +1/2$ at $d = d_I$, we can assume $\eta(\mathbf{r})$ described by the energy functional

$$\delta E[\epsilon_2, \eta] = a \int d\mathbf{r}\left[\left(\eta(\mathbf{r})^2 - \frac{1}{4}\right)^2 - g\left(\epsilon_2(\mathbf{r})^2 - \epsilon_{\text{IMT}}^2\right)\eta(\mathbf{r})\right], \tag{11}$$

with both $a$ and $g$ positive, where the potential well at $\eta < 0$ corresponds to smaller $d$ values, thus to the metal phase, while the well at $\eta > 0$ to the insulator with larger $d$, and both may coexist. The parameter $\epsilon_{\text{IMT}}^2$ plays the role of the above mentioned symmetry breaking term: $\epsilon_{\text{IMT}}^2 > 0$ ($\epsilon_{\text{IMT}}^2 < 0$) lowers the well centred at $\eta < 0$ ($\eta > 0$), thus stabilizes the metal (insulator). A finite shear strain contributes to $\epsilon_{\text{IMT}}^2$ favouring the insulating well at $\eta > 0$, thus the term $-g\epsilon_2(\mathbf{r})^2\eta(\mathbf{r})$ in Eq. (11).

Comparing with the phase diagram of $(Cr_xV_{1-x})_2O_3$ in Fig. 1 above the rhombohedral-monoclinic transition temperature, thus when $\epsilon_2^2 = 0$, we must conclude that $\epsilon_{\text{IMT}}^2$ is positive in the $\alpha$-phase and negative in the $\beta$ one, and thus decreases with increasing both $x$ and $T$, crossing zero at the first-order metal–insulator transition line $T_{\text{IMT}}(x)$.

Alternatively, $\epsilon_{\text{IMT}}^2$ can be regarded as the threshold strain above which an insulating phase becomes stable in a monoclinic phase with $\epsilon_2 > 0$.

**Total energy functional**. Adding the shear-strain energy $E[\boldsymbol{\epsilon}_2]$ in Eq. (6) to the dimer stretching contribution $\delta E[\epsilon_2, \eta]$ in Eq. (11), we obtain the total energy functional

$$E[\boldsymbol{\epsilon}_2, \eta] = E[\boldsymbol{\epsilon}_2] + \delta E[\epsilon_2, \eta], \tag{12}$$

which can describe rhombohedral and monoclinic phases, either metallic or insulating.

For instance, assuming $\epsilon_{\text{IMT}}^2 < 0$ from high to low temperatures in Eq. (11), the energy functional (12) predicts a transition from a rhombohedral insulator to a monoclinic one upon lowering $T$, i.e., the reduced temperature $\tau$, as indeed observed above 1% of Cr doping, see Fig. 1.

Pure $V_2O_3$ corresponds instead to assuming $\epsilon_{\text{IMT}}^2 > 0$ from high to low temperatures. In this case, the stable rhombohedral phase, with $\epsilon_2 = 0$, is metallic. Upon crossing the first-order structural transition at $T = T_c$, $\epsilon_2$ jumps directly to a finite value, $\epsilon_2(T_c)$, which grows upon further lowering $T$. If we assume, in agreement with the most recent experimental claims[53], that at equilibrium there is no monoclinic metal phase in between the rhombohedral metal and monoclinic insulator, we must conclude that just after the transition to the monoclinic phase $\epsilon_2(T_c)^2 - \epsilon_{\text{IMT}}^2 > 0$, so that the global monoclinic minimum is always insulating. Consequently, we fixed the parameters of the energy functional so that, assuming homogeneous phases, i.e., neglecting the Ginzburg term and the long-range potential, $K = \kappa = 0$ in Eq. (6), the phase diagram, see Fig. 4a, shows a direct first-order transition from a corundum metal to a monoclinic insulator, with a coexistence region of width $\Delta T \simeq 40$K consistent with experiments. We emphasize that the absence of a stable monoclinic metal does not exclude its presence as a metastable phase that is allowed by the energy functional (12), and which we indeed find, see Fig. 4b.

**Domains at equilibrium**. We mentioned that the long-range elastic potential $\hat{U}(\mathbf{r})$ in (6) favours[64] the existence of domains in the lower symmetry monoclinic phase, which seem to persist also within the insulator-metal coexistence region across the first-order transition[3,32,52,53]. Since the configurational entropy in presence of different domains plays an important role at finite temperature, we cannot simply search for the minima of the classical energy functional Eq. (12) varying the reduced temperature $\tau$, as in Fig. 4, but

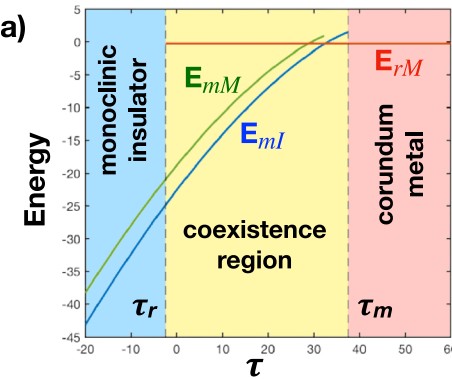
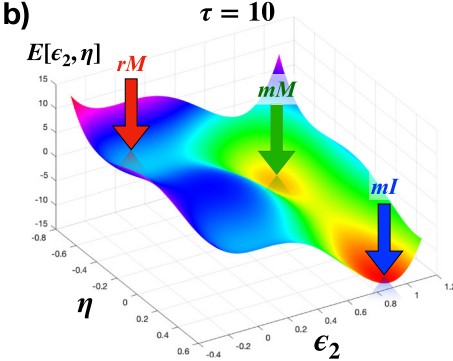

**Fig. 4 Phase diagram of uniform phases. a** Phase diagram from the energy functional (12) as a function of $\tau$ at $K = \kappa = 0$. $E_{rM}$, $E_{mM}$ and $E_{ml}$ are, respectively, the energies of the rhombohedral metal, red line, monoclinic metal, green line, and monoclinic insulator, blue line, namely, the depths of the corresponding local minima. The energy crossing between $E_{rM}$ and $E_{ml}$ signals the actual first-order transition. The vertical dashed lines at $\tau = \tau_m$ and $\tau = \tau_r$ are, respectively, the monoclinic and rhombohedral spinodal points. For $\tau \in [\tau_r, \tau_m]$ there is phase coexistence. We note the existence of a metastable monoclinic metal, with energy $E_{mM}$. **b** The energy landscape at $\tau = 10$ as a function of $\eta$ and $\epsilon_2 \geq 0$. We note the existence of three minima: a global monoclinic insulating one (*ml*), and two local minima: a lower monoclinic metal (*mM*), at $\epsilon_2 > 0$ and $\eta < 0$, and an upper rhombohedral metal (*rM*), at $\epsilon_2 = 0$ and $\eta < 0$.

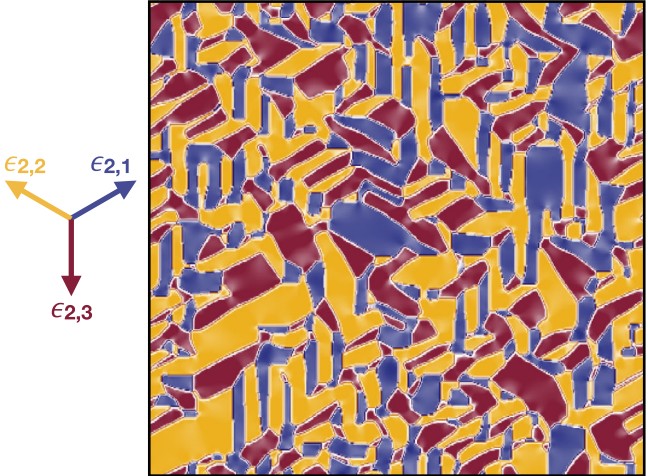

**Fig. 5 Calculated nanotexture of the monoclinic insulating phase.** Calculated real space distribution of the shear strain $\epsilon_2(\mathbf{r})$ deep inside the monoclinic insulating phase. The colours correspond to the three equivalent shear-strain vectors $\epsilon_{2,1} = (+\sqrt{3}/2, +1/2)$, $\epsilon_{2,2} = (-\sqrt{3}/2, +1/2)$ and $\epsilon_{2,3} = (0, -1)$, shown on the left, which characterize the three equivalent monoclinic twins, see Eq. (5). The figure is a superposition of three different ones. The first is obtained plotting the *y*-component of $\epsilon_2(\mathbf{r})$ on a colour scale from $-1$ (plum) to 0 (white); the second plotting the *x*-component from $+\sqrt{3}/2$ (blue) to 0 (white), and the third plotting still the *x*-component but now from $-\sqrt{3}/2$ (orange gold) to 0 (white). Evidently, when the lighter regions of all three plots overlap that implies both *x* and *y* components are nearly zero, thus a small strain. We note that the interfaces between different domains evidently satisfy the curl-free condition (9).

we need to calculate actual thermodynamic averages. For that, we take inspiration from the mean-field theory developed in refs. [66,75], which was originally developed for ferroelastic transitions, and we extend it to treat the lattice and electronic insulator-to-metal nanotextured dynamics in a Mott material. We discuss thoroughly such mean-field scheme in Supplementary Note 4 (See the Supplementary Material), while here we just present the results.

In Fig. 5 we show the calculated real space distribution of the shear strain $\epsilon_2(\mathbf{r})$ at low temperature, i.e., deep inside the

monoclinic insulator. As expected, the distribution is not homogeneous, but shows coexistence of equivalent monoclinic twins, each characterized by a colour that corresponds to one of the three equivalent shear-strain vectors $\epsilon_{2,i}$, $i = 1, 2, 3$, see Eq. (5), which are shown on the left in Fig. 5. We note that the interface between two domains, i.e., two strain vectors $\epsilon_{2,i}$ and $\epsilon_{2,j}$, $i \neq j$, is directed along the third vector, $\epsilon_{2,k}$, $k \neq i, j$, in accordance with the curl-free condition (9), and with the experimental data presented in the section "Spontaneous nanotexture of the monoclinic insulating phase". In addition, the strain along the interfaces is strongly suppressed as compared to the interior of each domain, as indicated by the lighter regions in Fig. 5. The inherent suppression of the strain amplitude at the domain boundaries will turn out to play a fundamental role in seeding and stabilizing the photo-induced non-thermal metallic phase.

In the top panel of Fig. 6, we show the calculated real-space distribution of $\epsilon_2(\mathbf{r})$ within the monoclinic-rhombohedral coexistence region across the temperature-driven first-order transition, where the green colour indicates the rhombohedral domains. The bottom panel of Fig. 6 instead shows the real-space distribution of $\epsilon_2(\mathbf{r})^2 = \epsilon_2(\mathbf{r}) \cdot \epsilon_2(\mathbf{r})$, which is finite in any monoclinic domain, and zero in rhombohedral ones. We mentioned that the insulator is locally stable if the shear-strain amplitude square $\epsilon_2(\mathbf{r})^2 > \epsilon_{IMT}^2$, otherwise the locally stable phase is metallic. For that reason, we use in the bottom panel of Fig. 6 a blue colorscale for all regions where $\epsilon_2(\mathbf{r})^2 > \epsilon_{IMT}^2$, and a red colorscale for $\epsilon_2(\mathbf{r})^2 < \epsilon_{IMT}^2$, the two colours thus distinguishing between insulating and metallic domains. We note that, as $T$ raises, metallic domains start to nucleate first with a residual monoclinic strain, light red, that soon disappears, dark red. This gives evidence that the metastable monoclinic metal does appear across the monoclinic-rhombohedral phase transition, even though it gives in to the rhombohedral metal before a percolating metal cluster first sets in, in accordance with experiments[53].

We also observe that rhombohedral domains, dark red, have triangular shapes, as dictated by the curl-free condition (9) at the interfaces between monoclinic and rhombohedral domains. This pattern does not resemble the observed experimental one[3]. This difference is due to the *c*-plane orientation that we use, in contrast to the A-plane one in experiment[3].

**Space-dependent non-equilibrium model.** We note that the above theoretical modelling may also account for the experimental

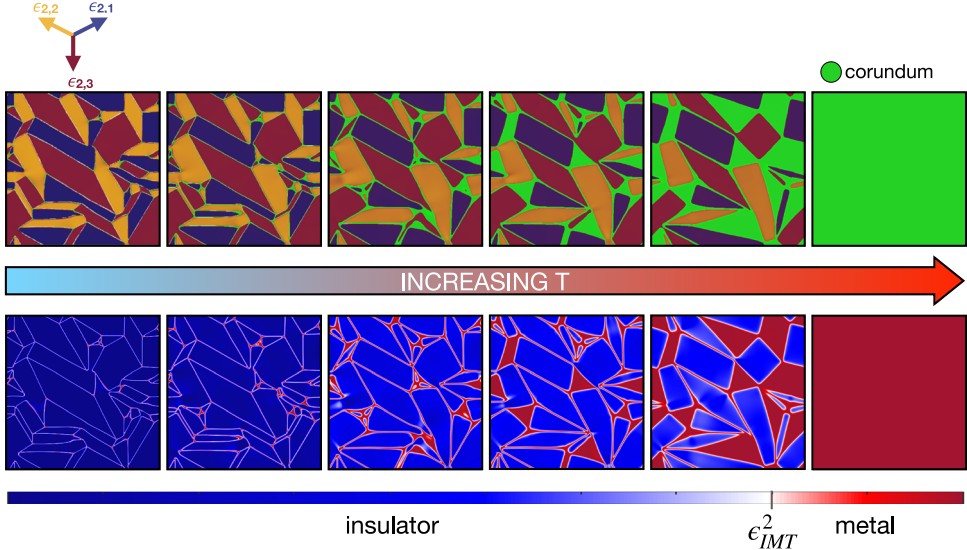

**Fig. 6 Simulated dynamics of monoclinic domains across the transition.** Calculated domain pattern across the first-order transition upon rising temperature. Top panel: colour map of the shear strain $\epsilon_2(\mathbf{r})$, specifically, the monoclinic domains are indicated by the same colour scale as that used in Fig. 5, in which the colours indicate the three equivalent shear-strain vectors. The rhombohedral domains are indicated in green. To enhance the contrast, we use, unlike Fig. 5, a colour scale that does not distinguish between small and zero strain. Bottom panel: colour map of the square modulus of the shear strain $\epsilon_2(\mathbf{r})^2$. Blue keys indicate the monoclinic insulating domains, $\epsilon_2(\mathbf{r})^2 > \epsilon_{IMT}^2$, while red keys the metallic ones, $\epsilon_2(\mathbf{r})^2 < \epsilon_{IMT}^2$.

evidences[10,20,32] of a photo-induced metal phase in V$_2$O$_3$. As discussed in refs. [20,32], the main effect of the 1.5 eV laser pulse is to transfer electrons from the $e_g^\pi$ to the $a_{1g}$ derived bands, see Fig. 1c. The increase in $a_{1g}$ population at the expense of the $e_g^\pi$ one is supposed to yield a transient reduction of the actual trigonal field splitting between $e_g^\pi$ and $a_{1g}$ orbitals. Indeed, an effect of Coulomb repulsion, captured already by mean-field approximation, is to make occupied and empty state repel each other, thus increasing their energy separation. This is the reason why the trigonal field splitting between lower $e_g^\pi$ and upper $a_{1g}$ is renormalised upward by Coulomb interaction. Evidently, also the reverse holds true: if electrons are transferred from the more occupied $e_g^\pi$ state to the less occupied $a_{1g}$ one, the energy separation between them diminishes, which amounts to a net reduction of the effective trigonal field splitting. Such effect of the laser pump can be easily included in the double-well potential (11) that describes the dimer stretching, i.e., the trigonal splitting, by adding a laser fluence, $f$, dependent term linear in $\eta$, namely

$$\delta E[\epsilon_2, \eta] \rightarrow \delta E[\epsilon_2, \eta] + \int d\mathbf{r}\,\mu(f)\,\eta(\mathbf{r}), \quad (13)$$

with $\mu(f) > 0$, being zero at $f = 0$ and growing with it, thus favouring the metal state with $\eta < 0$. This term is actually equivalent to a fluence-dependent threshold strain

$$\epsilon_{IMT}^2 \rightarrow \epsilon_{IMT}^2(f) \equiv \epsilon_{IMT}^2 + \mu(f)/g, \quad (14)$$

that increases with $f$. Looking at the bottom panel of Fig. 6, such upward shift of $\epsilon_{IMT}(f)$ implies not only that formerly insulating regions with $\epsilon_{IMT} < \epsilon_2(\mathbf{r}) < \epsilon_{IMT}(f)$ may turn metallic, but also the possibility that a laser pulse with a fluence exceeding a threshold value stabilizes the formerly metastable monoclinic metal. Indeed, we earlier mentioned that we fix $\epsilon_{IMT}$ smaller than the shear-strain amplitude $\epsilon_2(T_c)$ just after the rhombohedral to monoclinic first-order transition, which ensures the absence of a stable monoclinic metal at equilibrium. After laser irradiation, $\epsilon_{IMT}(f)$ may well surpass the shear-strain amplitude, in turn reduced by heating effects, when $f$ exceeds a threshold fluence, thus stabilizing the monoclinic metal, metastable at equilibrium.

In order to simulate the spatial dynamics of the laser-induced metalization, we start from the calculated shear-strain map, reported in Fig. 5. As the pump excitation fluence rises, the concurrent increase of $\epsilon_{IMT}(f)$ leads to the possible nucleation of non-thermal metal regions with finite monoclinic shear strain, whenever the condition $\epsilon_2(\mathbf{r}) \leq \epsilon_{IMT}(f)$ is met. In Fig. 7 we report the spatial configuration of such domains (purple areas), which are metallic and yet characterized by the same in-plane monoclinic nanotexture of the insulating phase (see Supplementary Note 4 (See the Supplementary Material) for the parameters). We note that the non-thermal metal starts nucleating at the boundaries between different monoclinic twins, where, as previously discussed, the strain is constrained to smaller values than in the interior of each domain. As $\epsilon_{IMT}(f)$ increases, the filling fraction of the non-thermal metal phase grows progressively up to the point of occupying the entire region.

**Time-resolved PEEM experiments**. To demonstrate that such intriguing scenario indeed realizes in photoexcited V$_2$O$_3$, we developed a novel time-resolved X-ray PEEM experiment (see Fig. 3a) with 30 nm and 80 ps spatial and temporal resolution (see "Methods" and Supplementary Note 1 (See the Supplementary Material) for the experimental details). With this imaging method we studied the temporal response of the monoclinic domains triggered by a properly synchronized pulsed laser excitation (1.5 eV photon energy; ~50 fs pulse duration) capable of impulsively changing the $e_g^\pi$ and $a_{1g}$ band population and possibly inducing the non-thermal metallic monoclinic state. At this photon energy, the light penetration depth is ~300 nm, as extracted from the refractive index reported in ref. [76]. Considering that the escape depth of electrons in the total yield configuration used for acquiring the PEEM images is of the order of 3–5 nm[77,78], the pump excitation can be assumed as homogeneous within the probed volume.

The time-resolved experiments were performed on a 50 nm V$_2$O$_3$ crystalline film deposited by oxygen-assisted molecular beam epitaxy on a sapphire substrate, with the $c$-axis perpendicular to the surface[40]. In order to characterize the IMT, we first measured the temperature-dependent optical properties at a

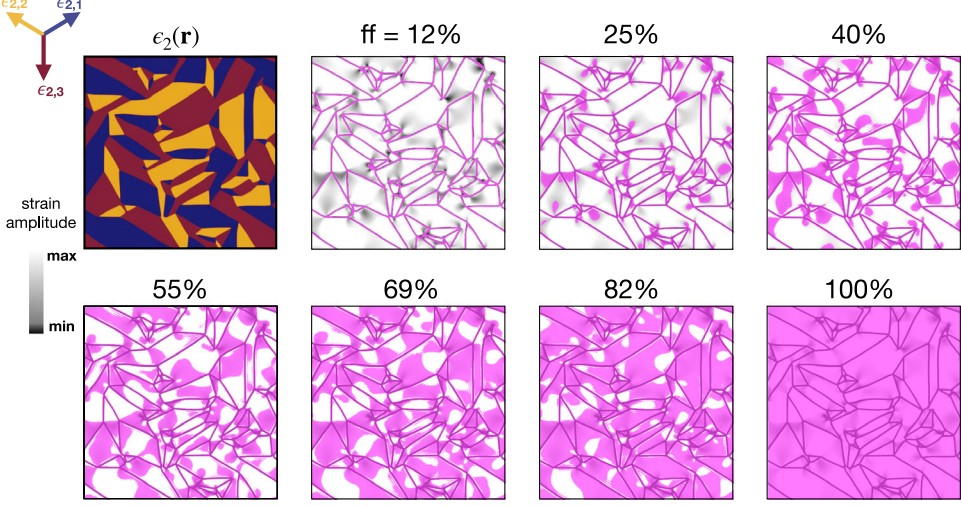

**Fig. 7 Metastable monoclinic metal.** Map of the space-dependent calculated shear-strain amplitude (grey colour-scale) and metastable monoclinic metallic regions (purple solid areas). The different filling fractions (ff) of the metastable phase correspond to different values of $\epsilon_{IMT}(f)$. Darker grey indicates smaller strain amplitude. The purple areas highlight the spatial regions in which the electronic metallic solution with monoclinic strain is the stable one (absolute minimum in the free-energy), i.e. when the condition $\epsilon_2(\mathbf{r}) \leq \epsilon_{IMT}(f)$ is fulfilled. We stress that in all panels the same equilibrium space-dependent monoclinic shear strain, $\epsilon_2(\mathbf{r})$, of the top-left panel (colours correspond to the three equivalent shear-strain vectors, as in Fig. 5) is considered. The filling fraction of the photo-induced non-thermal metallic phase is indicated for each panel.

selected probe photon energy (2.4 eV) during the heating and cooling cycles. The curve reported in Fig. 8a shows the typical hysteresis of the insulator to metal transition with mid-point at $T_c \simeq 140$ K, slightly smaller than that observed in bulk crystals as a consequence of the film residual strain[40]. The reflectivity at 2.4 eV drops by 14% when the temperature is increased from 100 K (insulating phase) to 180 K (metallic phase), while the film resistivity drops by approximately 3 orders of magnitude (see Supplementary Fig. 5 (See the Supplementary Material)). Figure 8b, c show equilibrium XLD-PEEM images of the sample taken at T=100 K and 180 K, respectively. As discussed at length in Sec. "Spontaneous nanotexture of the monoclinic insulating phase" the XLD-PEEM images clearly evidence in the low-temperature monoclinic phase the formation of stripe-like domains corresponding to different monoclinic twins[32]. When the temperature is increased well above $T_c$, the monoclinic nanotexture is replaced by a homogeneous corundum phase with an almost absent XLD contrast.

As extensively discussed in the literature[20,24,25,32,51,79], the electronic IMT can be also photo-induced by using ultrashort infrared pulses as the external control parameter. When the excitation is intense enough, the insulating phase collapses on a timescale of ~30–50 ps transforming into a new phase with the same optical properties as the metallic one, both in the THz[24,25] and in the infrared/visible[32] frequency range. We monitored such transformation in our sample by recording the relative reflectivity variation after 100 ps between the optical pump and probe, i.e. when the time-resolved signal already reached a plateau (see Fig. S6). In Fig. 8d we show the relative reflectivity variation at 2.4 eV probe photon energy as a function of the impinging pump fluence. Above ≈ 8mJ/cm² , the measured reflectivity drop perfectly matches the equilibrium reflectivity difference between the insulating and metallic phases, thus demonstrating that the whole of the pumped volume is turned into the electronic metallic phase.

In order to investigate the dynamics of the monoclinic domains during the photo-induced insulator-to-metal transition, we performed a XLD-PEEM experiment exploiting the inherent

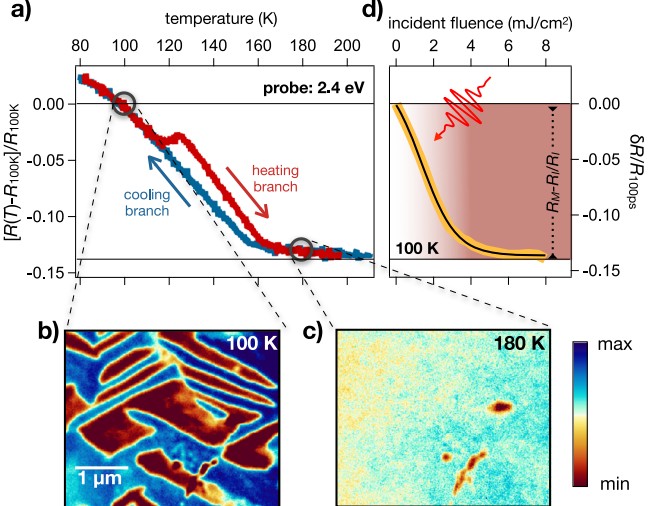

**Fig. 8 Insulator-to-metal transition. a** Reflectivity change of the $V_2O_3$ crystal across the temperature-driven insulator-to-metal phase transition. The sample reflectivity is measured at 2.4 eV photon energy as a function of the sample temperature during the heating (red curve) and cooling (blue curve) processes. The graph displays the relative reflectivity variation with respect to the reflectivity measured at T = 100 K. **b** PEEM image taken at 100 K evidencing stripe-like domains corresponding to the different monoclinic distortions. Note that the experimental configuration of the image shown (polarization parallel to one of the hexagon edges) is such that only two domains are visible. **c** PEEM image taken at 180 K evidencing a homogeneous background, typical of the metallic corundum phase. The colour scale indicates the amplitude of the PEEM signal. **d** The asymptotic value of the relative reflectivity variation (yellow trace), i.e. $\delta R/R$(100 ps) = [R($\Delta t$ = 100 ps)-R($\Delta t$ = 0 ps)]/R($\Delta t$ = 0 ps) where $\Delta t$ is the pump–probe delay, is measured at 2.4 eV probe photon energy and T = 100 K as a function of the incident pump fluence. The black solid line is a guide to the eye.

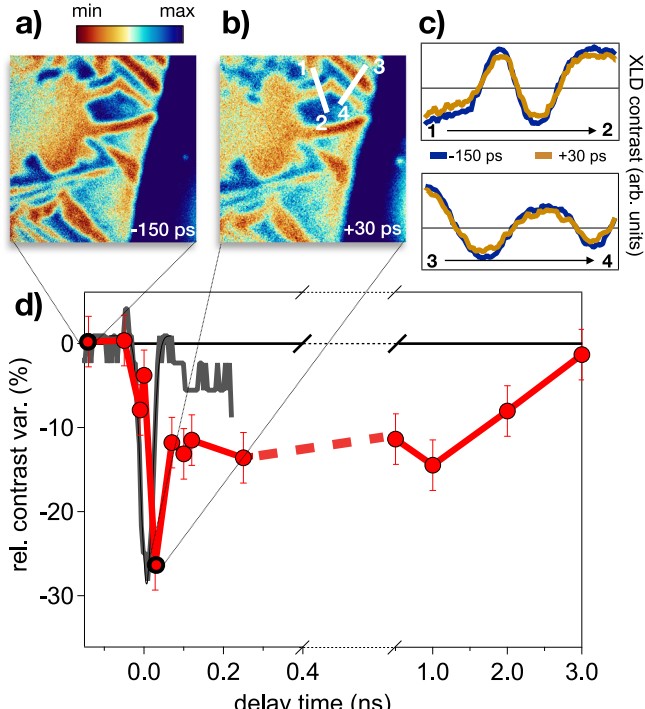

**Fig. 9 Time-resolved microscopy. a** Time-resolved PEEM image taken at $T$ = 100 K and at negative delay (−150 ps) between the infrared pump and the X-ray probe pulses. **b** Time-resolved PEEM image taken at $T$ = 100 K and at positive delay (+30 ps) between the infrared pump and the X-ray probe pulses. The colour scale for both panels (**a**) and (**b**) is the same than that used in Fig. 8. **c** XLD contrast profile along the segments 1 → 2 and 3 → 4, as indicated in panel (**b**). The blue profiles are taken at negative delay (−150 ps), whereas the yellow profiles correspond to positive delay (+30 ps). **d** Relative contrast (see the Supplementary Information) between different domains as a function of the delay between the infrared pump and the X-ray probe pulses. The error bar accounts for the average fluctuation of the signal within the domains considered for the calculation of the relative contrast. The grey solid line is the cross-correlation between the infrared pump and the X-ray probe pulses measured by exploiting the non-linear photoemission from surface impurities on the sample (see Supplementary Note 1 (See the Supplementary Material)).

pulse structure of the synchrotron X-ray radiation and the synchronization with a femtosecond laser source, which allows turning the XLD-PEEM experiment into a time-resolved microscopy tool with 80 ps time-resolution (see "Methods" and Supplementary Note 1 (See the Supplementary Material)). The experiment was carried out on the same sample, and in the same experimental conditions as the optical pump–probe results reported in Fig. 8d, in order to avoid possible artefacts related to different average heating in the two experiments[26]. We underline that after heating and cooling the system, the topology of the domains remains unchanged. The repeatability and stability of the domain formation, discussed in ref. [32], is the prerequisite for performing the time-resolved experiment, which consists of an average over many different pulses. The pump–probe spatial and temporal overlap, as well as the pump spot size, were carefully checked by exploiting the non-linear photoemission from surface impurities, as explained in detail in Supplementary Note 1 (See the Supplementary Material), and by imaging the pump beam at the sample position.

Figure 9 displays a typical image of the monoclinic domains 150 ps before (panel a) and 30 ps after (panel b) the excitation with laser pulses at 22 ± 4 mJ/cm² fluence, which exceeds by far

the threshold necessary to photo-induce the complete transformation into the electronic metallic phase. Although the low signal-to-noise and the large background signals accumulated during the long acquisition times make a detailed analysis of the space-dependent dynamics very difficult, it is clear that the topology of the monoclinic domains remains almost unchanged after the excitation. It is however instructive to compare local photo-induced changes of the XLD-PEEM signal along specific lines. In Fig. 9c we report the signal profiles along two selected lines, which perpendicularly cut some of the monoclinic stripe-like domains. The comparison between the profiles at negative and positive delays demonstrates a weak and almost uniform suppression of the contrast between the signal originating from different domains corresponding to monoclinic distortion along different directions. To better analyze the long-time dynamics as a function of the time delay between the infrared pump and the X-ray probe, in Fig. 9d we report the relative variation of the contrast of the signal from different monoclinic domains (red and blue regions), as obtained by integrating over different areas of the PEEM image (see Section S1). The average XLD contrast decreases by almost 30% within 50 ps from the excitation and reaches a plateau, corresponding to ~10% variation, in the 100 ps–1 ns time span. The PEEM signal fully recovers within 3 ns, which corresponds to the cooling time of the sample[10]. No signature of long-time melting of the monoclinic nanotexture is observed.

Although the contrast between the XLD signal from different monoclinic domains is mainly originated by the directionality of the $t_{2g}$ orbitals, as discussed in Section "Spontaneous nanotexture of the monoclinic insulating phase" and S4, a weak contribution to the X-ray absorption signal is also given by the orbital occupation of the initial state of the $L_{2,3}$ transition. The cluster multiplet calculations reported in ref. [41] show that the in-plane absorption at 520 eV significantly changes if fully polarized different initial states are considered. In particular, the difference between X-ray absorption at 520 eV and 518 eV, which is used as the reference to cancel backgrounds, decreases by ~50% if the orbital occupation changes from $e_g^\pi e_g^\pi$ to $e_g^\pi a_{1g}$. The observed transient decrease of the XLD contrast between neighbouring domains is thus compatible with the creation of a metastable metallic state with enhanced $a_{1g}$ occupation and the same in-plane monoclinic distortion of the low-temperature insulating phase. We also note that, although the pump-induced variation of the $e_g^\pi$ and $a_{1g}$ occupations is very rapid, the growth of metastable metallic domains takes place on much longer timescales. More specifically, the increase of the $a_{1g}$ occupation in the metastable monoclinic metallic phase implies the restoring of the vanadium dimer length of the corundum structure. This process is much slower since it involves the rearrangement of the vanadium dimers over distances of the order of the typical size of the monoclinic domains (~250 nm). This latter structural transformation acts as the bottleneck for the proliferation of the non-thermal phase transformation[32].

A possible route to control the photo-induced non-thermal metallic phase is given by interface strain engineering[80], which allows to control the residual interface strain in the $V_2O_3$ film. Quite naturally, the presence of residual tensile strain in the film may enhance $\epsilon_{IMT}$ and favour the emergence of monoclinic metallic regions. In Fig. 10a we compare the fluence-dependent filling fraction measured on the $V_2O_3$ film used for tr-PEEM measurements to that obtained on a similar $V_2O_3$ film in which the residual strain is diminished by means of a $Cr_2O_3$ buffer layer[40] (see "Methods"). The fluence-dependent data show remarkably different metallization dynamics, compatible with a decrease of $\epsilon_{IMT}$ in the film with the $Cr_2O_3$ buffer layer. We underline that, in both cases, the morphology of monoclinic domains is very similar,

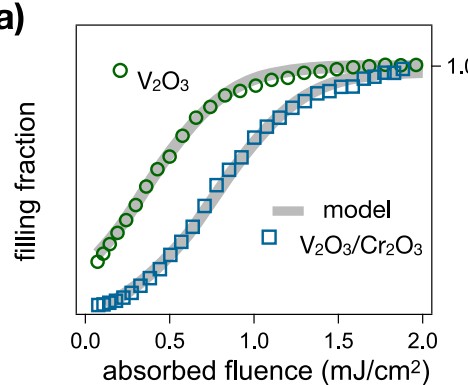

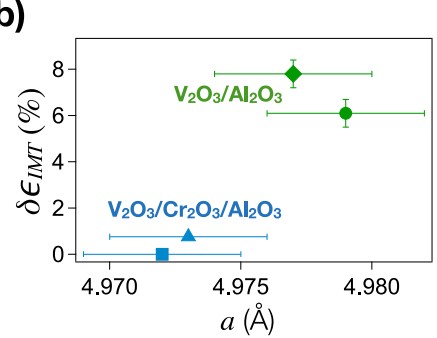

**Fig. 10 Strain engineering of the metastable monoclinic metallic phase.**
**a** Metallic filling fraction, retrieved from the asymptotic value of the relative reflectivity variation, i.e. $\delta R/R(100 \text{ ps})$, as a function of the pump incident fluence for a 50 nm $V_2O_3$ film directly grown on the sapphire substrate (green circles) and for a 55 nm $V_2O_3$ film grown on a 60 nm $Cr_2O_3$ buffer layer (blue squares). The grey solid lines represent the numerical filling fractions, calculated as the ratio between non-thermal metallic areas (purple areas in Fig. 7) and the total area. **b** Values of the estimated critical strain variation $\delta\epsilon_{IMT}$, calculated with respect to the reference sample $V_2O_3/Cr_2O_3/Al_2O_3$ (55 nm/60 nm/substrate). $\delta\epsilon_{IMT}$ is plotted as a function of the room temperature $a$-axis lattice parameter, as measured by X-ray diffraction, for samples with (blue points) and without (green points) the $Cr_2O_3$ buffer layer. The symbols refer to the following samples: blue square $V_2O_3/Cr_2O_3/Al_2O_3$ (55 nm/60 nm/substrate); blue triangle $V_2O_3/Cr_2O_3/Al_2O_3$ (67 nm/40 nm/substrate); green diamond $V_2O_3/Al_2O_3$ (40 nm/substrate); green circle $V_2O_3/Al_2O_3$ (50 nm/substrate). The error bars account for the uncertainty in the measurement of the lattice parameter and in the determination of $\delta\epsilon_{IMT}$.

as shown by PEEM images (see Fig. S7). The difference in the $\epsilon_{IMT}$ values does not impact on the monoclinic nanotexture, which is governed by the functional (6), but rather controls the fragility toward the emergence of the photo-induced non-thermal metallic phase. In Fig. 10b we present more data points showing the correlation between residual tensile strain and the value of $\epsilon_{IMT}$.

In this work, we have developed a coarse-grained model, based on the minimization of a Landau–Ginzburg energy functional, to account for the space-dependent lattice and electronic dynamics across the insulator-to-metal transition in the archetypal Mott insulator $V_2O_3$. The spontaneous long-range nanotexture, originated from the minimization of the lattice energy, emerges as a key element to describe both the temperature-driven and the photo-induced transition. The reduced-strain regions at the domain boundaries and corners provide the necessary template for the nucleation of metallic domains. In out-of-equilibrium conditions, the domain boundaries stabilize and protect the photo-induced non-thermal monoclinic metallic state, which would be unstable at equilibrium and in homogeneous systems.

Although the reported theory and experiments refer to the Mott transition in $V_2O_3$, the present results unveil a profound and general link between the real-space topology, the transition dynamics and the emergence of non-thermal electronic states in quantum materials. The combination of multiscale modelling and microscopy experiments with time-resolution offers new platforms to understand and control the transition dynamics of solids which exhibit spontaneous self-organization at the nanoscale. Indeed, the complexity of space-dependent solid-solid phase transformations involving different degrees of freedom (electrons, lattice, spins, etc.) opens new exciting possibilities for achieving the full control of the transition and for synthesizing novel emerging metastable states that do not exist at equilibrium and in homogeneous phases. The present results suggest possible routes to control metastable metallicity via the topology of the nanotexture. For example, crystals cut along different directions should exhibit little or no texture at all, which could change the dynamics of the photo-induced phase. On the other hand, the improvement in time-resolution and the use of free-electron-laser-based spatially resolved experiments are expected to provide fundamental information about the early time dynamics of non-thermal metallization, which is missed in the present experiments. The combination of real-space morphology control via interface engineering, electric fields or pressure, with the development of novel excitation schemes to coherently manipulate insulator to metal phase transitions[81] are expected to open new routes for achieving the full and reversible control of the electronic properties of correlated oxides.

From the theoretical side, our results call for the development of realistic models that capture the long-range dynamics and the complexity of electronic transitions, which are usually tackled starting from a microscopic approach. On the one hand, theory should provide a guide to calculate the X-ray absorption signals of non-thermal phases, which do not exist at equilibrium. This would support the next generation of experiments in addressing the actual electronic and lattice configurations of transient metastable states. On the other hand, microscopic theory could help in linking the lattice parameters ad residual strain to the phenomenological control parameters, e.g. $\epsilon_{IMT}$, which enter into the Landau–Ginzburg description. This effort would provide new keys to engineer the intrinsic nanotexture and control non-thermal phases.

In conclusion, the present results justify the ongoing efforts to develop novel table-top and large-scale facility time-resolved microscopy techniques to investigate the intertwining between non-thermal properties and real-space morphology in quantum correlated materials[82]. Addressing the role of real space inhomogeneities and intrinsic strain nanotexture will be crucial to finally clarify the long-standing issue about the possibility of fully decoupling and control the electronic and structural phase transitions in vanadates[3,20,22,23,26,51–53,83] and other Mott materials.

## Methods

**Experimental setup.** Time-resolved PEEM measurements have been performed at the I06 beamline at the Diamond Light Source synchrotron. In order to carry out the experiment, the synchrotron was set to the hybrid injection pattern, in which a higher charge single-bunch is separated by the smaller charge multibunches by a 150 ns time gap. By gating the detector to avoid the multibunches it is possible to use the gap in the injection pattern to perform pump–probe experiment (see Fig. 1b, main text). The frequency of the X-ray single pulses is 533.8 kHz while the temporal width is 80 ps, which sets the temporal resolution of the experiment. The X-ray pulses are synchronized to a pump laser source delivering $\simeq$ 50 fs pulses at 1.5 eV photon energy and 26.7 kHz rep. rate. The photoemission signal relative to the X-ray pulses synchronized to the pump laser is acquired by proper electronic gating. The pump–probe delay is controlled by electronically modifying the opto-acoustic mirror of the laser cavity.

**Samples**. An epitaxial $V_2O_3$ film with thickness $d = 50$ nm is deposited by oxygen-assisted molecular beam epitaxy (MBE) in a vacuum chamber with a base pressure of $10^{-9}$ Torr. The (0001)-$Al_2O_3$ substrate is used without prior cleaning and is slowly heated to a growth temperature of 700°. Vanadium is evaporated from an electron gun with a deposition rate of 0.1 Å/s, and an oxygen partial pressure of $6.2 \times 10^{-6}$ Torr is used during the growth[40]. Under these conditions, a single crystalline film with the $c$-axis oriented perpendicular to the surface is obtained. To facilitate the spatial overlap between the optical pump and the X-ray probe we nanopatterned markers constituted by 40 nm Au/5 nm Ti thick layers deposited on the sample surface. The residual strain at the $Al_2O_3/V_2O_3$ interface induces an expansion of the in-plane lattice parameter $a$, which changes from 4.954 Å (bulk) to $\simeq 4.978$ Å (film), as determined from X-ray diffraction. The use of a $Cr_2O_3$ buffer layer (lattice parameter $a = 4.9528$ Å) relaxes the tensile strain in the $V_2O_3$ film by 0.12% (lattice parameter $a \simeq 4.972$ Å in $V_2O_3$ for the $Al_2O_3/Cr_2O_3/V_2O_3$ configuration).

## Data availability

The data generated in this study are available at http://hdl.handle.net/10807/208360.

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

## Acknowledgements

C.G., A.R. and P.F. acknowledge financial support from MIUR through the PRIN 2015 (Prot. 2015C5SEJJ001) and PRIN 2017 (Prot. 20172H2SC4_005) programs. C.G. and G.F. acknowledge support from Università Cattolica del Sacro Cuore through D.1, D.2.2 and D.3.1 grants. We acknowledge Diamond Light Source for the provision of beamtime under proposals number SI18897 and MM21700. J.-P.L. acknowledges financial support by the KU Leuven Research Funds, Project No. KAC24/18/056, No. C14/17/080 and iBOF/21/084 as well as the Research Funds of the INTERREG-E-TEST Project (EMR113) and INTERREG-VL-NL-ETPATHFINDER Project (0559). M.M. acknowledges support from "Severo Ochoa" Programme for Centres of Excellence in R&D (MINCINN, Grant SEV-2016-0686). M.F. has received funding from the European Research Council (ERC) under the European Union's Horizon 2020 research and innovation programme, Grant agreement No. 692670 "FIRSTORM".

## Author contributions

A.R., P.F., P.H., A.F., F.M., S.S.D., M.M., J.-P.L. and C.G. conceived the project and carried out the time-resolved experiments at Diamond Light Source (UK). C.G. coordinated the research activities with input from all the coauthors, particularly A.R., P.F., F.M., M.M., J.-P.L. and M.F. A.R., P.F., M.M, J.-P.L. and C.G. analyzed the data. A.R., P.F., G.F. F.B. and C.G. developed the time-resolved setup for time-resolved reflectivity experiments. P.H., M.M. and J.-P- Locquet performed the MBE thin film growth experiments and X-ray diffraction characterization and analysis. All the authors participated in the discussion of the results and contributed to the revision of the manuscript. M.F. developed the theoretical framework with main inputs from A.DP. and C.G. A.DP. performed numerical calculations based on Landau–Ginzburg functionals. A.R., M.F. and C.G. drafted the first version of the manuscript.

## Competing interests

The authors declare no competing interests.
