## [Peer Review File · Nature Communications]

REVIEWER COMMENTS

Reviewer #1 (Remarks to the Author):

The authors report on a transient metallic state realized exciting the monoclinic antiferromagnetic phase of V₂O₃ out of equilibrium.

Topic-wise this work is definitely interesting and of potential broad impact. The experimental technique as well as the theoretical analysis used are state-of-the-art and of high quality. Before recommending publication in Nat. Commun. I find that the points below should be addressed. In my opinion, they would indeed improve the manuscript and the discussion, in particular related to the previous literature on V₂O₃.

1) As the authors discuss in the introduction, Cr-doped V₂O₃ undergoes a temperature- as well as pressure-driven first-order Mott transition. This, in addition to the original Refs. 31-34, has been also revisited in recent works:

Phys. Rev. B 61, 11506–11509 (2000) DOI 10.1103/PhysRevB.61.11506

Phys. Rev. Lett. 104, 047401 (2010) DOI 10.1103/PhysRevLett.104.047401

Phys. Status Solidi B, 1–14 (2013) DOI 10.1002/pssb.201248476

Phys. Rev. B 90, 115115 (2014) DOI 10.1103/PhysRevB.90.115115

Even though this transition and its first-order properties have been described so far mostly from the point of view of the electronic degrees of freedom, the structural role of Cr-dopants turns out to be important. This is touched upon in these references, that focus on the dramatic effects induced by small changes of the lattice structure within the paramagnetic, non-long-range ordered corundum phase of this compound.

The present manuscript describes a first-order phenomenon (for pure V₂O₃) related instead to a structural and symmetry change as well as to long-range magnetic order.

Can the author discuss in the introduction and/or in the final part of the manuscript a bit more deeply the comparison with the above-mentioned temperature-/pressure-driven paramagnetic Mott transition? For instance, is the role of strong local correlations crucial in their treatment? In turn, can the Landau-Ginzburg approach introduced here be extended to investigate the paramagnetic Mott

transition in Cr-doped V₂O₃, where also structural (though not-symmetry-lowering) strain is important, as also mentioned in the references above?

2) Phase separation - In addition to Ref. 3 there are important references not considered by the authors, reporting spatial manifestations of the phase separation accompanying the paramagnetic Mott transition:

Phys. Rev. Lett. 97, 195502 (2006) DOI 10.1103/PhysRevLett.97.195502

Nature Commun. 1, 105 (2010) DOI 10.1038/ncomms1109

Adding a related discussion could be useful: One question is for instance, whether the phenomenon illustrated in Fig. 12 is reversible or not. Does the pattern and the position remain those of Fig. 12 by repeating several times the experiment on the same region of the film? Could that be also influenced and/or stabilized by defects in the epitaxial growth?

3) In the case of thin films grown on Al₂O₃, even with the Cr₂O₃ buffer, the lattice parameters are strongly modified w.r.t. the bulk values, as the authors discuss also in the supplementary material. In Ref. 31 (main text) the ratio between the c and a lattice constants is shown to be key to understand the metallic properties of V₂O₃. As this ratio is modified, e.g. because of temperature [see Phys. Rev. B 77, 113107 (2008) DOI 10.1103/PhysRevB.77.113107] or strain [Appl. Phys. Lett. 107, 241901 (2015); DOI 10.1063/1.4937456], the relative weight of the so-called "alpha" and "beta" structures is influenced. Further, related to point 1), can the authors comment whether they could broaden their modus operandi (both theoretical and experimental) to discuss and, possibly, induce, the paramagnetic Mott transition involving the "alpha" and "beta" phases?

4) I suggest the authors to characterize a bit more the metallic phase reached by pumping. Is this possible based on the information they have? Should one think of this phase roughly as the metal of the corundum phase above T_N or is it totally different? Should one expect that in the Cr-doped

case a similar experiment as the present one would give a Mott insulator after bringing the compound out of equilibrium and suppressing the antiferromagnetic order?

5) Related to point 4, the authors measure the reflectivity at 2.4 eV but can one infer anything that is happening closer to the Fermi level? In Ref. 32, for pure V₂O₃, a resistivity drop of six orders of magnitude is reported. Can one compare this to Fig. S5?

Reviewer #2 (Remarks to the Author):

This joint experimental and theoretical work focuses on equilibrium and light-induced metastable phases of the prototypical Mott insulator V₂O₃. By parametrizing the decisive features of this compound, shear strain and dimer length, the authors introduce a sophisticated coarse-grained Ginzburg-Landau model to capture the real-space lattice and electron structure of the insulator-to-metal transition and identify a metastable metallic phase. This is complemented by optical reflectivity and time-resolved photoemission electron microscopy (trPEEM) measurements, demonstrating the existence of the metastable nonequilibrium phase predicted by the model.

This work is novel and addresses a mechanism for optical control of strongly correlated materials, a research field of rapidly growing interest. Furthermore, this work explores the nonequilibrium phase diagram of vanadium oxides, which is highly debated. The manuscript is well written and the overall structure is logical.

However, before supporting publication, I have a number of comments that need to be addressed. While the detailed discussion of the background is helpful, the clarity of the presentation does not match the high standard of Nature Communications (e.g. grouping the large number of small figure into fewer ones, see below). Furthermore, a number of assumptions of the theoretical model need to be clarified, and the experimental work lacks a clear description, which makes it difficult to assess the

validity of the observed metastable phase. Detailed comments follow.

1. How do the authors justify the assumption that dimer tilting and elongation can be treated as separate degrees of freedom? Furthermore, how do the authors justify the parametrization of the insulator-to-metal transition purely by dimer bond length, particularly in view of recent ARPES work, see Lo Vecchio, et al., PRL 117, 166401 (2016), that questions the validity of the existing explanation of the IMT transition by correlation-enhanced crystal field splitting?

2. The authors argue that the optical excitation modifies the population of a_{1g} and e_g orbitals, which reduces the trigonal field splitting, leading to a transient metallization. In the model, the optical excitation is treated similar to dimer bond stretching (404ff), despite a different physical origin. If the mechanism leading to the metastable state is indeed purely electronic, the switching timescales are expected on the order of 100fs, as shown by Ref. 20, rather than the observed 30-50 ps (line 455). What limits the timescale of switching to the metastable state and how does the proposed mechanism relate to the observed timescales? Can the authors discuss how their theoretical modelling and experimental observation relate to experimental evidence of a previously reported photoinduced metallic state with significantly shorter buildup times and lifetimes (Ref. 20)?

3. The non-equilibrium model suggests a modulation of the strain within the domains (see Figure 10, grey color code). If this metastable state is actually reached in the experiment, this modification of strain should also be apparent in the trPEEM images. Is this actually reflected in the experimental data?

4. The experimental data in Figs. 12-14 discusses a large range of fluences. Can the authors explain how the fluences have been determined and comment on the geometries of the different experiments? Determining the laser spot profile from surface defects can only yield a very rough approximation due to the non-linearity of the signal. Did the authors crosscheck this method of determining the spot size? Why does the fluence behavior vary drastically between the optical measurements presented in Fig. 12d (saturation at $4\text{mJ}/\text{cm}^2$) and 14a (saturation at $1\text{mJ}/\text{cm}^2$)? A comparison of absorbed fluences would support the claim of the authors that trPEEM and optical experiments probe the same metastable phase. Is there any independent observable in the PEEM measurements (e.g. in the energy spectrum) that confirms the presence of a transient metallic state?

5. What exactly is encoded in the optical reflectivity measurements? Is the reflectivity signal purely related to the electronic Mott-insulator transition, or is it also affected by strain? How robust is the assignment of transient metallicity from the optical reflectivity?

6. The discussion of the energy functional (equations 13 and 14) and its dependence on η and ϵ is difficult to follow. A visualization of the functional for critical values of the relevant parameters could be helpful and should be added to a revised manuscript. E.g., in the phase coexistence region τ within $[\tau_r, \tau_m]$, where is the global minimum of the functional?

Some minor points:

a. Can the authors estimate the lifetime of the photoinduced metallic state? Does the system fully relax to the ground state between subsequent pulses?

b. An extensive discussion of the background is given in section 2 with too many individual figures. Grouping related figures will increase clarity.

c. The central aspect of the multiscale modelling (section 3), the parametrization using 2 distinct, coupled degrees of freedom to describe the structural and electronic transition, gets lost in an exhaustive discussion of the background. To improve readability, I suggest to clarify the theoretical approach and the parametrization of dimer tilting and dimer elongation early

on.

d. In the abstract (line 37), the authors state that their results can be readily extended to many families of Mott insulating materials. As the applied model appears quite specific to V₂O₃, I suggest to tone down this statement.

e. Can the authors clarify what they mean by “messing up a Mott transition” (line 57f)?

f. Figure 3 is confusing and does not convey what actually drives the Mott-to-metal transition. How does this situation e.g. map onto a Hubbard-like model and its phase diagram?

g. In line 290, the authors state that τ encodes the electronic effects. Can the authors comment on where this does arise from?

h. How is the data in Figure 6a obtained and what is plotted? I suggest to include a color scale in Fig. 6a and indicate the 3 symbols in all panels. In Figure 6b, how are the intensities of the different domains extracted? Did the authors account for the apparent shift of the measured sector for different polarizations? Can the authors indicate the absolute scale in Fig. 6b? Specifically, if Fig. 6a and b show both XLD-signals, as indicated in the caption, one would expect the same signal level for the pink cursor position at 0 deg and 90 deg from Fig. 6b.

This, however is apparently not the case in the left and right images of Fig. 6a (orange vs. blue color).

i. In Figure 10, what is the control parameter? Does ε_{IMT} change for the different panels? Furthermore, the strain amplitude is encoded in grey and does vary within the domains. However, the figure caption also states that the same monoclinic shear strain is retained in all panels. How do these contradictory statements fit together?

j. The authors state that nucleation of metallic domains starts with monoclinic strain (lines 393ff). However, in Figure 9, I can only identify metallic regions (red) with corundum structure (green). Can the authors indicate which regions they refer to by light red?

k. In caption of Figure 11, the term 'X-ray detector' is used. Are the authors detecting X-rays or electrons?

l. Why do the temperature-dependent reflectivity curve (Fig. 12a) and the resistivity curve (Fig. S5) not feature a clear jump, as one expects for a 1st order transition?

m. The statement in line 206f is unclear. How do the considerations above enforce a perpendicular domain interface?

n. In equation 9, the parameter K is not defined. In equation 16, g is not defined.

o. The variable γ is used for multiple quantities (equation 3 and 9).

p. Several spelling mistakes, e.g., line 187 demonstrates, line 471 photoemision, line 505 an in homogeneous, line 511 sold-solid

q. References 20 and 29 are identical

r. Several erroneous references to figures throughout the manuscript and supplement, e.g.

- reference to Supp. Fig. S3 in line 459,
- reference to Fig S4 in line 492
- several references in the last paragraph on page 3 of the Supplement

Reviewer #3 (Remarks to the Author):

In this work the authors present a three-fold achievement: 1) High quality V2O3 films are grown with verifiable twin domain configurations that are quite unique to this growth orientation; the XLD-PEEM imaging results are spectacular. 2) A unique multi-scale model is developed to give explicit understanding of the interplay between accommodation strain and stability of native insulator and metal phases, with ramifications for intermediary states. 3) Time-resolved nano-imaging shows conclusively that even a film "fully switched" to non-equilibrium metallic phase remains solidly in a monoclinic configuration. For these findings, this work is likely to influence and help guide future studies of stabilized intermediary and non-equilibrium phases in Mott systems. Therefore I recommend publication in Nature Communications, but first subject to some major revisions for clarification and improvement, as I elaborate below:

I can broadly state the two most important problems that should be addressed: 1) The central interpretation of this work posits that twin domain boundaries provide a pathway to metastable phases not achieved in equilibrium. This conclusion is supported by their Ginzburg-Landau formulation, but only insofar as the energy functional is well motivated. In this regard, while the authors' formulation of the strain contribution is exceptional and likely to set precedent for treatments of this material, coupling to the "electronic" order parameter is scarcely detailed and may be misconstrued as ad hoc. The authors should motivate this part of their formulation in better detail, particularly in relation to Takana's work [ref 47], to ensure confidence in their predictions. 2) The time-resolved PEEM results are spectacular and the capability to resolve any real-space features emerging after pump excitation should be given more attention and discussion. The authors seem quick to conclude "the pattern is unchanged," and leave it at that. While this leads to accord with their interpretation, readers like myself are eager to know what new real-space information can be seen or not seen as a consequence of pumping. Origins for transient impacts on the XLD contrast

should be given more discussion, and some speculation is welcome. Future experimental directions building on this work should be discussed in the conclusion.

I now specify smaller points of concern, which nevertheless should be addressed before the manuscript is suitable for publication in Nature Communications:

- Grammatical errors or misspellings should be corrected, e.g. "ephasise" [sic], and colloquialisms like "mess up the Mott transition" (line 57) should be avoided. The authors are also free to select their own title, but the subtext "a richness in disguise" might be substituted for something more concretely descriptive, like "a pathway to metastable metallicity" or similar.

- Ref 35 and 5 appear to be duplicates.

- The number of distinct figures appears excessive and will be problematic for typesetting. Figs. 1-5 might be combined in one figure since they conceptually describe the V₂O₃ structure and transition. I suggest later where other figures may be combined.

- Discussion of x,y,z axes requires these be labeled in Fig. 2

- Fig. 4: Red/blue for spin orientation should be labeled in a legend, because casual readers comparing Figs. 1-5 with Fig. S1 would be confused that the same colors also indicate V-dimer displacement. Also, the "rotation of dimers" about the b_m axis should be labeled here. Something like the depiction in Fig. 2 of Tanaka would be clearer.

- Fig. 6: The position labels could be made more clear with e.g. better contrasting colors. This figure also demands a cartoon of the PEEM setup showing the angle of incidence of X-rays and associated plane where polarization is controllable. A related cartoon appears in a later figure; my forthcoming comment suggest to bring that graphic forward to this figure. "Arb. Units" on the y-axis of Fig. 6b should be made explicit. What is the overall depth of contrast permissible at the X-ray incidence angle?

- In Fig. 6a, the inferred pseudo-hexagonal axes should also be labeled for one or more domains, to show how they align with the domain orientations. It is stated that "the interface between two of the three monoclinic twins is perpendicular to the b_m axis of the third one", can that be visualized here? Would it be less cryptic to state that the "habit plane" between two twins is actually the complement of their two "local" y-axes, along which their primary distortions form?

- Line 238 states: "Such weak-coupling Fermi surface instability has its strong coupling counterpart." What is meant by this? Are weak- and strong-coupling regimes realized at different levels of theory, or at different regimes of the phase transition? Clarify.

- At line 253, and Eq. (4): The authors rightly decide "not to weigh down the text" with derivations of the Ginzburg-Landau theory, deferring those to the supplement, but since details of the shear strain definition are anyway found there, these might also be removed from the main text. It seems to me the sole message to convey in the main text is that order parameters comprise a 3-fold scalar shear

strain and a one-fold scalar dimer distance η (or d , why are two notations used for the same quantity?). If this understanding is correct, then the discussion can be streamlined; if it is incorrect, the authors should clarify.

- Line 304 refers the elastic constants which are provided context in the supplement, but not in the main text; here they should be labeled, or better yet their appearance should be deferred entirely to the supplement.

- The discussion of γ and τ in Eq. 9 and their compatibility with the bulk phase diagram is hard to follow. Presumably, coupling between shear strain and the intra-dimer distance d (again, $=\eta$?), together with explicit temperature dependence of the latter (e.g. via electronic entropic contribution), is sufficient to “drive” the structural transition. For a qualitative example, see for instance Paquet & Leroux-Hugon, “Electron correlations and electron-lattice interactions in the metal-insulator, ferroelastic transition in VO₂”, (1980). Instead, the authors attribute primary temperature dependence to the stiffness constants, which seems puzzling and only circumstantially supported by literature evidence. Another sufficient source of temperature dependence can be the phonon entropy and resultant temperature-dependent lattice constants, as shown by Han & Millis, “Lattice Energetics and Correlation-Driven Metal-Insulator Transitions: The Case of Ca₂RuO₄”, (2018). It might be understood that all these formal treatments can yield identical energy functionals in the strain when the dimer order parameter is “integrated out”, and in this sense the authors’ physical conclusions can remain robust. Nevertheless, this multi-scale model is relatively novel for V₂O₃ and may set precedent, so it is important that the microscopic interpretation be reported faithfully. Therefore the authors should reconsider their reported source of temperature dependence, or else, they must defend their present choice.

- The authors should explain more clearly where Eq. 4 comes from; is it a reproduction from the Tanaka’s work? The foregoing elaborate discussion elaborating on d and the shear strain as distinct order parameters is surprisingly absent at this crucial stage, where it might provide some intuition about their coupling. Can one understand the allowed coupling of ϵ and d (or η) based on symmetry of the distortion? For instance, one might suppose a bilinear coupling of d and shear strain is allowed.. Also, the stability of a monoclinic metal seems to owe mostly to the strength of the “double well” assigned to η , as compared to the strain coupling g . Thus, it is imperative to microscopically motivate this double well. In this way, the reader may be convinced the monoclinic metal is not merely an artifact of ad hoc selections for the energy functional.

- Line 361 reports, “We emphasise [sic] that the absence of a stable monoclinic metal does not exclude its presence as a metastable phase that is allowed by the energy functional (14), and which we indeed find, see Fig. 7.” However the trace in Fig. 7 only reports the energy of the monoclinic metal, but does not demonstrate it to be metastable. The authors should accompany this panel with plots of the free energy landscape in which the putative monoclinic metal locates. As an example, see Fig. 4C of Lee, Eom, et al. “Isostructural metal-insulator transition in VO₂” (2018), whose analogous treatment in VO₂ should also be cited.

- Fig. 8 demands a proper color scale to understand what is being plotted. There are evidently shear strain values in the presented map that interpolate between the three “clock points”, as indicated by the caption, “Lighter regions indicate a reduced strain amplitude.” What are the values? Moreover, the authors should explain how intermediary strain values emerge from the “mean field” treatment

described by the supplement, in which it is stated “ \mathbf{S}_n are unit length two-component vectors”. How then do non-unity strain values emerge from this model? Since these localized points at reduced strain are a key prediction and a fundamental result for this work, the authors must admit no ambiguity here. In addition, the authors should also describe how the volume integral (or momentum sum in Eq. S67) is handled in view of the quasi-two dimensional nature of the thin film. This treatment of dimensionality is expected to have ramifications for effective long-range interaction, and therefore for the resultant micro texture. For an example, see section 3 of the Supplementary Information for McLeod et al. “Multi-messenger nano-imaging of hidden magnetism in a manganite film” (2020).

- Can the authors comment on why the strain is suppressed at domain wall boundaries? One can imagine the geometric mean of strain vectors across two dissimilar twins must be lower in magnitude than either. If that picture is too naive and the true mechanism is nontrivial, the authors should elaborate on the underlying cause. Perhaps a line-cut displaying the position-dependent shear strain across a twin domain boundary would help.

- At line 393, “We note that, as T raises, metallic domains start to nucleate first with a residual monoclinic strain, light red, that soon disappears, dark red. This gives evidence that the metastable monoclinic metal does appear across the monoclinic-rhombohedral phase transition...” This I do not understand. The authors should have plotted η in the bottom panel but have instead plotted the shear strain, from which one is evidently supposed to infer the metallicity. The reader should not be expected to solve Eq. 13 themselves, the actual metallicity result should be plotted here.

- At line 399, the meaning of “the experiment” in the following statement is ambiguous: “This pattern does not resemble the observed experimental one [Ref 35]. This difference is due to the c -plane orientation that we use, in contrast to the A -plane one in the experiment [Ref 35].” Are the authors drawing a comparison to Ref. 35? If so, the referenced work used r -plane sapphire, not a -plane; please correct.

- The discussion at line 409 describes a cogent treatment of the laser pulse, but the description is clunky and difficult to read. Also, it is clear how driving a change in trigonal splitting (equivalently bond length) would drive a difference in orbital occupation. But the converse scenario is implied by laser pumping; the authors should elaborate how modifying orbital occupancy leads to enhanced trigonal splitting.

- Fig. 9 and 10 might be combined to a single figure. These calculations would be easier to reconcile if they considered the same equilibrium domain pattern; is there a reason they do not?

- As alluded earlier, the Fig. 11 cartoon should be combined into Fig. 1 to subsume the recommended graphic that would describe the XLD-PEEM setup.

- In Fig. 13, the authors have claimed “the monoclinic nanotexture is fully retained”, whereas I can notice some dissimilarities (however slight) or added structure in the comparison of pre- and post-pump images. This is not necessarily to the disservice of the interpretation, but the authors should comment on these differences where they appear. Upon encountering the experimental result, a reader is tempted to find evidence for “smearing” enhanced foremost at twin boundaries, as predicted by the simulations. From the experimental perspective, is there any possibility to locate

attributes of the monoclinic metal in the PEEM images, or is there no straightforward contrast mechanism? Reflections on this question and suggestions for next-generation experiments should be included in the conclusion.

- The authors should comment why time-resolved “smearing” is observable at all- is there any associated feature (or reduction) emerging in the shear strain maps corresponding to Fig. 10? Alternatively, would a monoclinic metal be expected to provide reduced XLD contrast? Lastly, a trivial possibility that should be addressed is where the pump excitation “melts” the top 20% of the film to the corundum phase, which might be sufficient to yield an entire reflectivity response if the optical penetration depth is equal to or less than (e.g.) 8 nm, and likewise reducing the PEEM contrast. This possibility runs counter to the presented interpretation. The thin film provides a best platform to avoid such effects, but the authors should suitably dismiss such possibilities with analysis in reference to relevant V₂O₃ optical constants. For comparison, the relevant penetration depth of the soft X-rays (or escape depth of photoelectrons) should also be indicated.

- The comparison with V₂O₃ on Cr₂O₃ is an impressive achievement, particularly since the resultant nanotextures are quite comparable to the pure V₂O₃ film. The authors should clarify the putative relationship between ϵ_{IMT} and the lattice constant a , and Fig. 14b should include a vertical trace for the “native” V₂O₃ lattice constant to which strained values should be compared. Were the time-resolved PEEM results similar for V₂O₃ on Cr₂O₃?

- Line 503 asserts, “... the domain boundaries stabilize and protect the photoinduced non-thermal monoclinic metallic state, which would be unstable at equilibrium in [sic] homogeneous systems.” This seems to be one of the important implications of a c-cut film which enables the twin texture. We might suppose films with other growth directions would have fewer (or no) twin domains. Can the authors point to evidence the photo-induced response of the c-cut film is therefore qualitatively different (e.g. lower pumping threshold) from films at other growth planes, which likely have fewer twins? The authors should comment on the possibility to “engineer” twin domains as a more general recipe for encouraging intermediate states - could the twin texture be made even more fine-grained and if so by what means? In what other systems besides V₂O₃ might twin-stabilized intermediary states be possible?

Reply to Reviewer #1

First of all, we thank the Reviewer for the positive assessment on our work and for the useful comments that helped us improving the manuscript. In what follows we answer point by point to those comments and list the corresponding changes in the revised version.

- 1 **Role of Cr doping** – *As the authors discuss in the introduction, Cr-doped V₂O₃ undergoes a temperature- as well as pressure-driven first-order Mott transition. This, in addition to the original Refs. 31-34, has been also revisited in recent works [...]. Even though this transition and its first-order properties have been described so far mostly from the point of view of the electronic degrees of freedom, the structural role of Cr-dopants turns out to be important. This is touched upon in these references, that focus on the dramatic effects induced by small changes of the lattice structure within the paramagnetic, non-long-range ordered corundum phase of this compound. The present manuscript describes a first-order phenomenon (for pure V₂O₃) related instead to a structural and symmetry change as well as to long-range magnetic order. Can the author discuss in the introduction and/or in the final part of the manuscript a bit more deeply the comparison with the above-mentioned temperature-/pressure-driven paramagnetic Mott transition? For instance, is the role of strong local correlations crucial in their treatment? In turn, can the Landau-Ginzburg approach introduced here be extended to investigate the paramagnetic Mott transition in Cr-doped V₂O₃, where also structural (though not-symmetry-lowering) strain is important, as also mentioned in the references above?*

As the Referee mentions, Cr doping in the rhombohedral phase has important structural effects: the hexagonal a_H lattice parameter grows while c_H diminishes, leading to a net $\gtrsim 1\%$ increase of the unit cell volume. In our notations, that corresponds to a positive strain component $\epsilon = (\epsilon_{11} + \epsilon_{22})/2$ overwhelming a negative $\epsilon_3 = \epsilon_{33}$. However, see, e.g., W.R. Robinson, *Acta Cryst.* **B31**, 1153 (1975) and S. Chen *et al.*, *J. Solid State Chem.* **44**, 192 (1982), the main structural change that characterises the paramagnetic insulating phase of $(\text{Cr}_x\text{V}_{1-x})_2\text{O}_3$, so-called β -phase, as compared to the metallic α -phase or to pure V_2O_3 is the substantial increase of the out-of-plane displacement of each V, i.e., the coordinate z that parametrises the vanadium Wyckoff position 12c in the $R\bar{3}c$ space group. The increase of z overcompensates the decrease of c_H yielding a longer dimer length $d = c_H(2z - 1/2)$. For instance, at room temperature and at 1% Cr-doping, d in the β -phase is 1.74% larger than in the α -phase, whereas a_H grows only by 0.87%. In other words, the increase of d seems to some extent unrelated to the volume expansion.

Therefore, it is undeniable that either crossing the corundum-metal to monoclinic-insulator transition lowering T , or the corundum-metal to corundum-insulator transition increasing Cr doping, or, finally, the α -to- β transition rising T , a sudden and substantial rise of the dimer length occurs, and that, we emphasise, despite the c -axis compression. Given

such one-to-one correspondence, and in the spirit of a Landau-Ginzburg approach, it is legitimate to promote the dimer length d to the rank of order parameter for the metal-insulator transition, which is what we do in the manuscript. That by no means implies that d drives the transition, which is ultimately electron-driven, but just that the behaviour of d faithfully mimics that of resistivity. In other words, the electronic degrees of freedom play a fundamental role in providing a double-well shape to the Born-Oppenheimer potential for d , consistent with d jumping at the transition, and that can be taken as representative of the metal-insulator transition. We mention that the β - α transition under pressure seems different from the metal-insulator transition driven by temperature or Cr doping, as reported in Phys. Rev. Lett. **104**, 047401 (2010) cited by the Reviewer. However, we are not aware of the behaviour of z across that pressure-driven transition, and therefore we cannot assess whether our description holds true even in that case, even though the evolution of the population imbalance $\Delta n = n_{e\pi} - n_{a1g}$ suggests it does not hold.

When writing the manuscript, we were a bit reluctant to discuss the α - β transition in light of the evidences of phase separation in the metal phase, also discussed in Phys. Status Solidi B **25**, 1251 (2013) cited by the Reviewer. However, following Reviewer's comment, we decided to add a discussion similar to the above one in the revised version of the manuscript, specifically when we argue why the dimer tilting and lengthening are two distinct degrees of freedom, and later when we justify the η -potential.

- 2 **Phase separation** – *In addition to Ref. 3 there are important references not considered by the authors, reporting spatial manifestations of the phase separation accompanying the paramagnetic Mott transition: Phys. Rev. Lett. 97, 195502 (2006); Nature Commun. 1, 105 (2010). Adding a related discussion could be useful: One question is for instance, whether the phenomenon illustrated in Fig. 12 is reversible or not. Does the pattern and the position remain those of Fig. 12 by repeating several times the experiment on the same region of the film? Could that be also influenced and/or stabilized by defects in the epitaxial growth?*

We thank the Reviewer for pointing to our attention those earlier works, which we now cite. The referee also raises another important and crucial point related to the reversibility and stability of the monoclinic nanotexture measured by PEEM at equilibrium. As previously reported in our earlier work [A. Ronchi et al. Phys. Rev. B **100**, 075111 (2019)] the nanopatterning is perfectly reproducible and monoclinic domains form exactly in the same position and with the same topology after heating and cooling cycles. Images proving this reversibility, which suggests pinning from defects and/or residual strain, are reported in Fig. 2 of the above reference. We also note that the pinning and reversibility of the domains is the necessary prerequisite to perform the time-resolved measurements reported in the present work. As any time-resolved experiment, the images are acquired with a stroboscopic technique, i.e. each image is the average of many different experiments at the same delay between the pump and probe pulses. If every time after heating or light

excitation the domains formed with different topology, it would be impossible to acquire any image like those reported in Figure 10 of the revised paper. In the revised version we have added a comment about reversibility of the domain formation, with reference to Fig. 2 of A. Ronchi et al. Phys. Rev. B **100**, 075111 (2019).

- 3 **$\alpha - \beta$ paramagnetic transition in thin films** – *In the case of thin films grown on Al₂O₃, even with the Cr₂O₃ buffer, the lattice parameters are strongly modified w.r.t. the bulk values, as the authors discuss also in the supplementary material. In Ref. 31 (main text) the ratio between the c and a lattice constants is shown to be key to understand the metallic properties of V₂O₃. As this ratio is modified, e.g. because of temperature [see Phys. Rev. B **77**, 113107 (2008)] or strain [Appl. Phys. Lett. **107**, 241901 (2015)], the relative weight of the so-called "alpha" and "beta" structures is influenced. Further, related to point 1), can the authors comment whether they could broaden their modus operandi (both theoretical and experimental) to discuss and, possibly, induce, the paramagnetic Mott transition involving the "alpha" and "beta" phases?*

The discussion in point 1) above gives a partial answer to this Reviewer's question. Indeed, as earlier mentioned, a key difference between those phases is the dimer length, longer in the β -phase than in the α one. Our Landau-Ginzburg functional includes a parameter ϵ_{IMT}^2 , see Eq. (13). As we mentioned, ϵ_{IMT}^2 must be negative for chromium doping when there is just a paramagnetic rhombohedral insulator to antiferromagnetic monoclinic insulator transition, and positive for pure V₂O₃. For intermediate doping levels x showing a $\alpha - \beta$ transition, we could imagine $\epsilon_{IMT}^2(x, T)$ dependent of x and T , specifically decreasing from positive to negative values with rising T above $T(x)$. That would lead to a first-order metal-to-insulator transition line at $T(x)$, which is accompanied by an upward jump of the dimer length and terminates in a second-order critical point, not in disagreement with experiments. We thank the Referee for rising this point. We added in the revised version a brief discussion on that issue.

As far as experiments are concerned, indeed we have in the pipeline further measurements on Cr-doped samples (see e.g. samples studied in Ref. P. Homm et al. APL Mater. **9**, 021116 (2021)) to address the formation of monoclinic nanotexture as a function of doping and residual strain. Definitive data on this subject will require an intense effort that goes beyond the current work.

- 4,5 **Non-thermal metal phase and Properties of the non-thermal phase at the Fermi level** – *I suggest the authors to characterize a bit more the metallic phase reached by pumping. Is this possible based on the information they have? Should one think of this phase roughly as the metal of the corundum phase above T_N or is it totally different? Should one expect that in the Cr-doped case a similar experiment as the present one would give a Mott insulator after bringing the compound out of equilibrium and suppressing the anti-ferromagnetic order?*

Related to point 4, the authors measure the reflectivity at 2.4 eV but can one infer anything that is happening closer to the Fermi level? In Ref. 32, for pure V2O3, a resistivity drop of six orders of magnitude is reported. Can one compare this to Fig. S5?

We merge the reply to the two last points raised by the Reviewer because they are strictly connected. Actually, the nature of the metastable metallic phase that is created by pumping is the main subject of this work. As far as the electronic properties are concerned, the data reported here, combined with the wealth of results already present in the literature, allows us to conclude that after pump excitation the full optical properties of the metallic corundum phase are transiently recovered. For sake of simplicity, in the present work we report just the reflectivity variation at a specific photon energy (2.4 eV), but the same results can be obtained at any wavelength in the visible range, as previously reported in Ref. A. Ronchi et al. Phys. Rev. B **100**, 075111 (2019), where we showed (see Fig. 4) that, after excitation, the reflectivity of the system is exactly equal to the reflectivity of the metallic corundum phase in the 1-2 eV energy range. Other pioneering works by colleagues [E. Abreu et al Phys. Rev. B **92**, 085130 (2015) and E. Abreu et al Phys. Rev. B **96**, 094309 (2017)] have shown that also the optical properties in the THz regime, i.e. the energy range more sensitive to the states close to the Fermi level and therefore to the metallization, transiently turn into those of the metallic phase. Whereas there is general agreement that pump excitation of the insulating monoclinic V₂O₃ results in a transient metallic phase from the point of view of the *electronic* properties, little is known about the lattice changes, particularly in films in which the spontaneous monoclinic nanotexture emerges. Our data (see Fig. 13) demonstrate that the in-plane monoclinic nanotexture is fully retained on the timescale in which the electronic metallic phase is photoinduced. For sake of completeness, we point out that the PEEM experiment, both in the equilibrium and time-resolved configurations, is just sensitive to the monoclinic in-plane distortion, whereas it is blind to possible variations of the out-of-plane distance of vanadium dimers. These results finally confirm the picture that we already proposed in Ref. A. Ronchi et al. Phys. Rev. B **100**, 075111 (2019) on the basis of speculations. Within this picture, the pump excitation triggers the rapid proliferation and growth of metallic regions that exhibit transient metal-like electronic properties and a nonthermal lattice structure, obtained by restoring the vanadium dimers distance of the corundum structure while retaining the monoclinic distortion of the vanadium hexagons. The final clarification of this complex non-thermal state will require the development of a multi probe time-resolved experiment sensitive, on the one hand, to the electronic properties and, on the other hand, to the full lattice configuration with nanometric spatial resolution. We hope that the ongoing efforts to develop time-resolved X-ray diffraction microscopy could help in this direction. In the revised version we have added further discussion of the transient metallic properties as emerging from THz and infrared-visible experiments reported in the literature.

We hope that the revised version and the discussion above satisfy the Reviewer, whom we

thanks again for her/his very helpful comments.

Reply to Reviewer #2

First of all, we thank the Reviewer for the positive assessment on our work and for the useful comments that helped us improving the manuscript. In what follows we answer point by point to those comments and list the corresponding changes in the revised version.

1. **Dimer tilting and elongation** – *How do the authors justify the assumption that dimer tilting and elongation can be treated as separate degrees of freedom? Furthermore, how do the authors justify the parametrization of the insulator-to-metal transition purely by dimer bond length, particularly in view of recent ARPES work, see Lo Vecchio, et al., PRL 117, 166401 (2016), that questions the validity of the existing explanation of the IMT transition by correlation-enhanced crystal field splitting?*

The in-plane and out-of-plane displacement of the vanadium atoms are different in nature: the former is responsible of the C_3 symmetry breaking, while the latter does not lower the space group symmetry but just increases the dimer length. To be more specific, the vanadium atoms in the $R\bar{3}c$ space group occupy the Wyckoff positions 12c that are characterised by a parameter z . The deviation $\delta = z - 1/3$, see Sect. S3 in the Supplementary Material, quantifies the displacement of V from the honeycomb plane, and thus the dimer length $d = c_H(1/6 + 2\delta)$, which is longer the bigger δ . Not to deal with the additional complications of the C_3 symmetry breaking, we can, e.g., compare the Cr-doped paramagnetic insulator with the metal, see, e.g., W.R. Robinson, *Acta Cryst.* **B31**, 1153 (1975) or S. Chen *et al.*, *J. Solid State Chem.* **44**, 192 (1982). This comparison shows that the main structural change in the insulator is the 15% increase of δ , corresponding to 1.73% increase of d , as opposed to a 0.87% increase of a_H and -0.46% decrease of c_H . This observation alone suggests that δ or, equivalently, the dimer length must be intimately correlated to the metal or insulator character of the sample, otherwise it would be rather strange that the dimer length increased across the metal-insulator transition despite the decrease of c_H .

Lo Vecchio *et al.* performed a remarkable ARPES experiment on V_2O_3 at $T = 200K$ in the metal phase, and found a_{1g} electron Fermi pockets coexisting with e_g^π hole ones. On the contrary, Poteryaev *et al.*, Ref. 44, calculated the band structure at $T = 390K$ by one-shot LDA+DMFT, and found just the a_{1g} electron Fermi pockets. Evidently, that calculation is not in agreement with the ARPES data. However, we believe that such

disagreement does not imply that the message of a correlation-enhanced trigonal field put forward by Poteryaev *et al.* is incorrect, but just that the calculated band structure is inaccurate. Indeed, as Lo Vecchio *et al.* instructively show in the Supplementary Material, band structure calculations in metal V_2O_3 critically depend on the DFT functional that is used, on the value of U in LDA or GGA plus U , on the way LDA+DMFT is implemented, etc...Therefore, it is rather subjective to trust one calculation instead of another. On the contrary, the fact that the e_g^π and a_{1g} centres of gravity are split more than straight LDA or GGA predict is common to all calculations, irrespective whether the top of the e_g^π band is found to be above or just below the Fermi energy. In fact, the correlation enhanced energy separation between lower e_g^π and upper a_{1g} is just the trivial consequence of the fact that interaction makes occupied and empty states repel each other; an effect that is well captured already by the Hartree-Fock approximation.

With that in mind, we believe, see, e.g., PRB B **87**, 205108 (2013) by one of the co-authors, that the Mott transition from the high temperature metal to the low temperature insulator occurs via the gradual emptying of the a_{1g} orbitals, which, if complete, would leave a half-filled e_g^π band prone to Mott's localisation and antiferromagnetic order. However, also because of the lattice involvement, the actual transition is pronouncedly first order and takes place before the a_{1g} fully empties. This scenario is not inconsistent with the data by Lo Vecchio *et al.*

For all the above reasons, we are convinced that the dimer length is indeed a valid indicator of the Mott transition in V_2O_3 . In the revised version, the above arguments are presented more clearly so to avoid any misunderstanding.

2. **Optical excitation process** – *The authors argue that the optical excitation modifies the population of a_{1g} and e_g orbitals, which reduces the trigonal field splitting, leading to a transient metallization. In the model, the optical excitation is treated similar to dimer bond stretching, despite a different physical origin. If the mechanism leading to the metastable state is indeed purely electronic, the switching timescales are expected on the order of 100fs, as shown by Ref. 20, rather than the observed 30-50 ps (line 455). What limits the timescale of switching to the metastable state and how does the proposed mechanism relate to the observed timescales? Can the authors discuss how their theoretical modelling and experimental observation relate to experimental evidence of a previously reported photoinduced metallic state with significantly shorter buildup times and lifetimes (Ref. 20)?*

We thank the Reviewer for raising this interesting point related to the timescales of the photoinduced phase transition. Some of the arguments have been already anticipated in the replies #4,5 to Reviewer 1. Actually, the physics reported in Ref. 20 is quite different since that paper just investigates the result of ultrafast pumping within the paramagnetic corundum insulating and metallic phases in Cr-doped samples, without inducing any insulator-to-metal transition starting from the low-temperature monoclinic insulating phase. The fast timescales that they observe just indicate that the pump indeed couples

to the a_{1g} and e_g^π populations, but this transient change of the band structure is rapidly (<1 ps) recovered and the systems comes back to the original state without any phase transition. Our case is much more complex because we start from the low-temperature monoclinic insulating phase and we end up in a metallic state photoinduced by the band excitation. The initial change of the a_{1g} and e_g^π occupation (<1 ps) is just the seed that triggers a complex multi-step transformation, which involves electronic and structural transformations on very different (and longer) timescales. It is well known from both optical [A. Ronchi et al. Phys. Rev. B **100**, 075111 (2019)] and THz [E. Abreu et al Phys. Rev. B **92**, 085130 (2015)] measurements that the insulator-to-metal transition completes on a timescale of ~ 50 ps. We report here the picture, already anticipated in reply to Reviewer 1, that emerges from the present experiments and the previously published results [A. Ronchi et al. Phys. Rev. B **100**, 075111 (2019)]. The pump-induced variation of the a_{1g} and e_g^π occupations (1 ps) triggers the rapid proliferation and growth of metallic regions that exhibit transient metal-like electronic properties and a nonthermal lattice structure, obtained by restoring the vanadium dimers distance of the corundum structure while retaining the monoclinic distortion of the vanadium hexagons. This process is much slower since it involves the rearrangement of the V dimers length over distances of the order of the typical size of the monoclinic domains (~ 250 nm). The initial formation of metallic seed takes place at the boundaries of the monoclinic domains. Considering the sound velocity of ~ 8 Km/s for V_2O_3 at 145 K [E. Abreu et al Phys. Rev. B **96**, 094309 (2017)], the time necessary for the growth of metallic seeds and the filling of the stripe-like monoclinic domains is of the order of 30 ps, well in agreement with the observed timescales. In the revised version of the manuscript we have added a more comprehensive description of this multi-step dynamics.

3. **Modulation of the strain** – *The non-equilibrium model suggests a modulation of the strain within the domains (see Figure 10, grey color code). If this metastable state is actually reached in the experiment, this modification of strain should also be apparent in the trPEEM images. Is this actually reflected in the experimental data?*

As correctly pointed out by the Reviewer, the strain is modulated within the domains and, more specifically, the strain amplitude decreases at the boundaries of the different monoclinic domains in order to fulfill the boundary conditions. This is however a property already of the *equilibrium* domains, as shown in Fig. 6 of the revised version, where the lighter colours indicate a reduced strain amplitude. The grey areas reported in Fig. 8 of the revised version (non equilibrium simulations) represent exactly the same strain amplitude as calculated from the equilibrium model and reported in Fig. 6. As stressed in the main text, the inherent reduced strain amplitude at the domain boundaries and corners is key to promote the photo-induced formation of the non-equilibrium metallic phase. Actually, a reduced XLD contrast is observed at the boundaries between adjacent domains in equilibrium PEEM experiments (see e.g. Fig. 4 of the revised version). However, the 30

nm spatial resolution of the technique tends to blur the domains boundaries, thus making a detailed analysis very difficult.

4. **Fluence dependence** – *The experimental data in Figs. 12-14 discusses a large range of fluences. Can the authors explain how the fluences have been determined and comment on the geometries of the different experiments? Determining the laser spot profile from surface defects can only yield a very rough approximation due to the non-linearity of the signal. Did the authors crosscheck this method of determining the spot size? Why does the fluence behavior vary drastically between the optical measurements presented in Fig. 12d (saturation at 4 mJ/cm²) and 14a (saturation at 1 mJ/cm²)? A comparison of absorbed fluences would support the claim of the authors that trPEEM and optical experiments probe the same metastable phase. Is there any independent observable in the PEEM measurements (e.g. in the energy spectrum) that confirms the presence of a transient metallic state?*

The Reviewer correctly focuses on the determination of the spot size in time-resolved experiments because it is crucial to determine the experimental fluence and compare different experiments. First of all, the referee is right in pointing out a possible inconsistency of the reported fluences in Figs. 12d and 14a of the previous version of the manuscript. The difference is related to the fact that in Fig. 12d (Fig. 9d in the revised version) we report the pump *incident* fluence, whereas in Fig. 14a (Fig. 11a in the revised version) we report the *absorbed* fluence to compare the actual excitation in two different samples (with and without the Cr₂O₃ buffer layer). Actually the data reported in Fig. 14a (Fig. 11a in the revised version) for the V₂O₃ film on sapphire are the same than those reported in Fig. 12d (Fig. 9d in the revised version), just rescaled to account for the sample absorption. We agree with the Reviewer that the two figures could be misleading, therefore in the revised version we specify in the labels whether we plot the absorbed or incident fluence.

The comparison between the two different experiments (time-resolved reflectivity and time-resolved PEEM) is definitely more tricky because we are comparing two different setups. In both the experiments we measured the same sample and in the same vacuum conditions. This allows us to rule out possible artifacts related to different heat dissipation in the two experiments, as recently carefully discussed in L. Vidas et al. Phys. Rev. X **10**, 031047 (2020). Therefore the main possible source of error in comparing the experimental fluences is the actual spot size, as pointed out by the referee. During the experiments we dedicated particular attention in determining the spot sizes on the sample and, consequently, the incident fluences. In the case of time-resolved reflectivity experiments, the spot size was carefully measured by both knife-edge techniques and by imaging of the pump beam at the sample position. For the time-resolved PEEM experiment, the experimental configuration is more complex because the sample is located in the PEEM chamber and the laser is focused through an optical window located outside the chamber. The laser spot size was monitored during the experiment by imaging the beam on a camera located at the same distance from the lens as the sample. However, to crosscheck this crucial param-

ter we also implemented the procedure described in the manuscript and Supplementary Information (see Fig. S4), which allowed us to retrieve the spot size dimension on the sample. The result of the *in-situ* measurement ($96 \times 110 \mu\text{m}^2$) perfectly matches the spot size measured on the camera via the imaging technique. The reported relative error bars on the spot size and the incident fluence (about 20% of the average value) accounts for the small uncertainties in determining the exact sample position in the UHV chamber. In the revised version of the Supplementary Information we added more details about the two independent measurements of the spot size.

Finally, the Reviewer asks for possible independent observation of transient metallicity in the time-resolved PEEM experiment. The PEEM signal is dominated by the dichroism originated by the monoclinic distortion and it is very difficult to disentangle it from the weaker signal originated solely from the electronic transition, which is associated to the dimer elongation and the change of the a_{1g} occupation. Nonetheless, the weak time-resolved signal reported in Fig. 10c of the revised version and corresponding to a transient decrease of the XLD contrast between different domains is compatible with both the calculations that we added at the end of section S4 and cluster multiplet calculations [Park et al. PRB 61, 11506 (2000)] that show a decrease of absorption at the $L_{2,3}$ edge as a consequence of the population change in the a_{1g} band. In the revised version of the manuscript we added a more comprehensive discussion of time-resolved data.

5. ***Optical reflectivity*** – *What exactly is encoded in the optical reflectivity measurements? Is the reflectivity signal purely related to the electronic Mott-insulator transition, or is it also affected by strain? How robust is the assignment of transient metallicity from the optical reflectivity?*

We addressed this point in the reply #4,5 to Reviewer 1. While in the present work we report for simplicity the transient reflectivity at 2.4 eV photon energy, similar results can be obtained in the whole infrared/visible range and in the THz domain. For example, in Ref. A. Ronchi et al. Phys. Rev. B **100**, 075111 (2019), we already showed (see Fig. 4) that the reflectivity of the photoinduced insulating phase is exactly equal to the reflectivity of the metallic corundum phase in the 1-2 eV energy range. Other colleagues [E. Abreu et al Phys. Rev. B **92**, 085130 (2015) and E. Abreu et al Phys. Rev. B **96**, 094309 (2017)] have shown that also the THz optical properties transiently turn into those of the metallic phase. The whole of the data already reported in the literature therefore demonstrates that pump excitation of the insulating monoclinic V_2O_3 results in a transient phase exhibiting the same *optical* properties of the metallic phase, from the THz to the visible. In the revised version we have added further discussion of the transient metallic properties as emerging from THz and infrared-visible experiments reported in the literature.

6. ***Minima of the Landau-Ginzburg functional*** – *The discussion of the energy functional (equations 13 and 14) and its dependence on η and ϵ is difficult to follow. A visualization*

of the functional for critical values of the relevant parameters could be helpful and should be added to a revised manuscript. E.g., in the phase coexistence region tau within $[\tau_r, \tau_m]$, where is the global minimum of the functional?

If we understand correctly, Reviewer's request is already addressed in Fig. 5 of the revised version, where we plot the energies of all minima of the Landau-Ginzburg functional versus the reduced temperature τ . In fact, those energies are just the values of the functional at the bottom of the corresponding potential well. Therefore, the global minimum corresponds to the lowest energy state, the other being just local minima. To make it clearer, we have extended the caption of that figure in the revised version, and added a panel with the free energy landscape at $\tau = 10$, where all minima are present.

We hope that the revised version and the discussion above satisfy the Reviewer, whom we thanks again for her/his very helpful comments.

Minor Points

a – *Can the authors estimate the lifetime of the photoinduced metallic state? Does the system fully relax to the ground state between subsequent pulses?*

As the referee suggests, the upper bound to the lifetime of the transient metallic state is given by the distance between subsequent pulses, i.e. about 40 μs in the time-resolved PEEM experiments. We performed also time-resolved reflectivity experiments at repetition rates as high as 400 kHz, which corresponds to 2.5 μs pulse distance, and still we didn't observe any pile up effect. The time-resolved PEEM data reported in Fig. 13 suggests that the lifetime is of the order of few nanoseconds.

b – *An extensive discussion of the background is given in section 2 with too many individual figures. Grouping related figures will increase clarity.*

We indeed regroup several figures, as suggested by the Reviewer.

c – *The central aspect of the multiscale modelling (section 3), the parametrization using 2 distinct, coupled degrees of freedom to describe the structural and electronic transition, gets lost in an exhaustive discussion of the background. To improve readability, I suggest to clarify the theoretical approach and the parametrization of dimer tilting and dimer elongation early on.*

We have revised the whole presentation of the theoretical modelling according to Reviewer's suggestions, as well as to the other Reviewers.

d – *In the abstract (line 37), the authors state that their results can be readily extended to many families of Mott insulating materials. As the applied model appears quite specific to V_2O_3 , I suggest to tone down this statement.*

We have revised that sentence.

e – *Can the authors clarify what they mean by messing up a Mott transition (line 57)?*

We turned the sentence into "Such circumstances might, at first sight, be regarded just as unwanted side effects that make the Mott transition more complex."

f – *Figure 3 is confusing and does not convey what actually drives the Mott-to-metal transition. How does this situation e.g. map onto a Hubbard-like model and its phase diagram?*

We better clarified that point in the caption of that figure, now Fig. 1(c). We mention that, while we can argue the overall evolution of the spectral weight approaching the metal-insulator transition, i.e., the gradual disappearance of the quasiparticle bands whose spectral weight is transferred to the Hubbard bands, fine details are still debated, as discussed in the work by Lo Vecchio *et al.* mentioned by the Reviewer.

g – *In line 290, the authors state that τ encodes the electronic effects. Can the authors comment on where this does arise from?*

In the revised version we better explain the meaning of the functional, which is actually the Born-Oppenheimer potential of the strain and dimer length and, as such, is substantially contributed by the electrons.

h – *How is the data in Figure 6a obtained and what is plotted? I suggest to include a color scale in Fig. 6a and indicate the 3 symbols in all panels. In Figure 6b, how are the intensities of the different domains extracted? Did the authors account for the apparent shift of the measured sector for different polarizations? Can the authors indicate the absolute scale in Fig. 6b? Specifically, if Fig. 6a and b show both XLD-signals, as indicated in the caption, one would expect the same signal level for the pink cursor position at 0 deg and 90 deg from Fig. 6b. This, however is apparently not the case in the left and right images of Fig. 6a (orange vs. blue color).*

The data plotted in Figure 4a of the revised manuscript report the PEEM intensity in arbitrary units. The color scale is exactly the same as the following figures 9 and 10 (revised version). We realized that we reported the colorscale only for Fig. 9. In the revised version, we added the colorscale to Fig. 4 and we indicated the symbols corresponding to the different monoclinic domains in the three panels. As far as Fig. 4b) is concerned, we added the zero line reference to help the reader. Actually the signal corresponding to the pink cursor is slightly positive for the first panel and slightly negative for the last one. This is the reason for the color difference in the two plots. In addition, in order to decrease the noise we had to integrate over a finite area centered on the cursors reported in panel a). This procedure increases the signal to noise ratio but tends to smear the contrast between different domains. In the revised version we increased the size of the cursors in order to make it more similar to the area of integration of the signal. We confirm that the small drift of the images has been considered in the analysis.

i – *In Figure 10, what is the control parameter? Does ϵ_{IMT} change for the different panels?*

Furthermore, the strain amplitude is encoded in grey and does vary within the domains. However, the figure caption also states that the same monoclinic shear strain is retained in all panels. How do these contradictory statements fit together?

Yes, the different panels report different filling fractions of the metastable monoclinic metallic phase obtained for different values of $\epsilon_{IMT}(f)$. The strain amplitude encoded in grey is spatially dependent and results from the calculation of $\epsilon_2(\mathbf{r})$ as reported in the text. However, $\epsilon_2(\mathbf{r})$ is the same for all the panels, whereas the only variable is $\epsilon_{IMT}(f)$. In the revised version we have clarified these issues.

- j – *The authors state that nucleation of metallic domains starts with monoclinic strain (lines 393). However, in Figure 9, I can only identify metallic regions (red) with corundum structure (green). Can the authors indicate which regions they refer to by light red?*

The Reviewer is right. In the top panel of figure 9 (Figure 6 of the revised version) we use a color scale that does not distinguish small strain from vanishing one: both of them are just green. We did that to better enhance the contrast because of the way we actually build that figure as well as the new figure 6 (former figure 9). For both of them, we had to face the problem of showing the space distribution of a two-component vector in a single colour plot. We solved it, maybe not in the best possible way, as we now present in detail inside the caption of Fig. 6. Specifically, figure 6 is actually a superposition of three different ones. Recalling that the three monoclinic twins correspond to $\epsilon_{2,1} = (+\sqrt{3}/2, +1/2)$, $\epsilon_{2,2} = (-\sqrt{3}/2, +1/2)$ and $\epsilon_{2,3} = (0, -1)$, the first figure is obtained plotting the y -component of $\epsilon_2(\mathbf{r})$ on a color scale from -1 (plum) to 0 (white); the second plotting the x -component from $+\sqrt{3}/2$ (blue) to 0 (white), and the third plotting still the x -component but now from $-\sqrt{3}/2$ (orange gold) to 0 (white). In that way, when the lighter regions of all three plots overlap that implies both x and y components are nearly zero, thus a small strain. In Fig. 6 that procedure works fine, since most of the sample has a strain equal to the maximum value. Approaching the transition, that procedure would yield to a very messy figure, which is the reason of our choice.

However that disputable choice is compensated by the bottom panel of Fig. 9, new figure 7, where we plot directly the modulus square of the strain vector, using the color scale shown below that panel. There, around all interfaces the color corresponds to a finite strain far from the extreme values and rather close to ϵ_{IMT}^2 . Moreover, in the small red regions of the first three panels, the red is evidently not as dark as in the last panel. For clarity, we mention that in the revised caption.

- k – *In caption of Figure 11, the term X-ray detector is used. Are the authors detecting X-rays or electrons?* We apologize for the mistake. Obviously we are detecting electrons! We fixed this issue.

- l – *Why do the temperature-dependent reflectivity curve (Fig. 12a) and the resistivity curve (Fig. S5) not feature a clear jump, as one expects for a 1st order transition?*

The hysteresis of the samples is rather large (50/60 K). The hysteresis region is characterised by spatial coexistence of phases and both the resistivity and reflectivity changes reflect this coexistence. Please, also note that the change of the electrical properties at the transition also overlaps with a more general temperature-dependence of the reflectivity and resistivity properties.

m – *The statement in line 206 is unclear. How do the considerations above enforce a perpendicular domain interface?*

The theoretical calculation of the x-ray linear dichroism in Sec. S4 (erroneously referred to as S5 in the original version) predicts that inside a single monoclinic domain the signal is maximum when the electric field is perpendicular to the monoclinic \mathbf{b}_m axis and minimum when it is parallel. The experimental data and the angle dependence of the signal, shown in the former figure 6, now figure 4, are consistent with the theoretic prediction only if the interface between two different twins is perpendicular to the \mathbf{b}_m axis of the third twin. We better explains how we get to that conclusion in the revised version, both in the text and in the caption of the new Fig. 4.

n – *In equation 9, the parameter K is not defined. In equation 16, g is not defined.*

We apologise for not having defined K , which is the strain stiffness, which we remedy in the revised version. The parameter g was actually defined in Eq. (13), Eq. (11) in the revised version.

o – *The variable γ is used for multiple quantities (equation 3 and 9).*

We changed the angle γ in Eq. (3) into γ_H , and correspondingly the other two angles into α_H and β_H , while the coupling constant that defines the functional remained γ .

p – *Several spelling mistakes, e.g., line 187 demonstrates, line 471 photoemision, line 505 an in homogeneous, line 511 sold-solid.*

We have corrected all these typos.

q – *References 20 and 29 are identical.*

We have fixed this issue.

r – *Several erroneous references to figures throughout the manuscript and supplement, e.g. reference to Supp. Fig. S3 in line 459, reference to Fig S4 in line 492, several references in the last paragraph on page 3 of the Supplement*

We have fixed these issues.

We hope that the above reply and the revised version satisfy the Reviewer, whom we sincerely thank for her/his fruitful comments.

Reply to Reviewer #3

First of all, we thank the Reviewer for the positive assessment on our work and for the useful comments that helped us improving the manuscript. The Reviewer raises two main issues, the first related to the Landau-Ginzburg formalism, and the second to the PEEM results. In addition, she/he raises several minor points. Some of them are closely related to the above two issues. For that reason, the reply that follows is organised in two parts: one concerns all Reviewer's comments about the theoretical content of the manuscript; the other the experimental results.

- 1 **Landau-Ginzburg formalism** – *The central interpretation of this work posits that twin domain boundaries provide a pathway to metastable phases not achieved in equilibrium. This conclusion is supported by their Ginzburg-Landau formulation, but only insofar as the energy functional is well motivated. In this regard, while the authors' formulation of the strain contribution is exceptional and likely to set precedent for treatments of this material, coupling to the "electronic" order parameter is scarcely detailed and may be misconstrued as ad hoc. The authors should motivate this part of their formulation in better detail, particularly in relation to Takana's work [ref 47], to ensure confidence in their predictions.*

We agree with the Referee that the η -functional in the original version was not well motivated. First of all, we gave for granted that η , i.e., the dimer length, is in one-to-one correspondence with the metal vs. insulator character. In addition, we assumed that the η field is controlled by a double-well potential $V(\eta)$.

An instructive insight into the first issue can be actually gained by comparing the structural parameters of the Cr-doped paramagnetic insulator, so-called β -phase, with those of the paramagnetic metal, α -phase, thus without the additional complications due to the monoclinic transformation. In the original version we omitted that comparison. However, also pushed by other Reviewers' comments, we realised such comparison is not only unavoidable, but also very significative.

The vanadium atoms in the $R\bar{3}c$ space group occupy the Wyckoff positions 12c that are characterised by a parameter z . The deviation $\delta = z - 1/3$, see Sect. S3 in the Supplementary Material, quantifies the vertical offset of V from the honeycomb plane, and thus the dimer length $d = c_H(1/6 + 2\delta)$, which is longer the bigger δ . The most noticeable structural change that occurs in the paramagnetic insulator, see, e.g., W.R. Robinson, *Acta Cryst.* **B31**, 1153 (1975) or S. Chen *et al.*, *J. Solid State Chem.* **44**, 192 (1982), is the 15% increase of δ , corresponding to 1.73% increase of d despite the -0.46% decrease of c_H , as opposed to a lower 0.87% increase of a_H . This observation suggests that δ or, equivalently, the dimer length must be intimately correlated to the metal versus insulator character of the sample. Therefore, δ can be taken as faithful indicator of the electronic

metal-insulator transition and, in the spirit of a Landau-Ginzburg formalism, it can be promoted to the order parameter η of that transition. That evidently implies that all parameters in the potential $V(\eta)$ are in reality strongly affected by the electronic degrees of freedom.

Let us move to the second issue: why $V(\eta)$ is a double-well potential. We note that, since across the $\alpha \rightarrow \beta$ first order transition δ jumps upward, one must conclude that the dimer length d has two equilibrium values, the smaller and the bigger characteristic of the metal and the insulator, respectively. Therefore, it naturally follows that η does feel a double-well potential. As in any first order phase transition, the control parameter is the relative depths of the two wells, which in our case is the parameter ϵ_{IMT}^2 : when positive it favours the metal phase, otherwise the insulator. Indeed, in absence of monoclinic distortion, and adding a Ginzburg term $\propto -\eta(\mathbf{r}) \nabla^2 \eta(\mathbf{r})$, the η -functional Eq. (13) describes a conventional liquid-gas transition. If we assume that $\epsilon_{IMT}^2(T, x)$ depends on temperature T and Cr concentration x , and decreases with rising T , then $T(x)$ such that $\epsilon_{IMT}^2(T(x), x) = 0$ defines the first order line that should terminate into a second order critical point. This modelling is consistent with the phase diagram of paramagnetic Cr doped V_2O_3 , and has been largely used to discuss its critical behaviour, see Limelette *et al.*, Science **302**, 89 (2003), as proposed, e.g., by Castellani *et al.*, Phys. Rev. Lett. **43**, 1957 (1979). Therefore, we believe that our choice of $V(\eta)$ is reasonable, and well motivated by experiments.

The next question is how the dimer length is affected by the dimer tilting of the monoclinic structure. Here, one can guess the result even without any calculation. Indeed, the tilting corresponds to the vanadium moving towards one of the three close-by octahedral voids. Evidently, that process can be completed only if the dimer also stretches a bit, as sketched in the figure 1. Therefore, it is rather natural to expect that the tilting, thus the finite shear strain ϵ_2 , favours a longer dimer length, and thus tends to lower the potential-well describing the insulator phase. This is exactly what we implement in our modelling. We also note that the coupling term $g \epsilon_2^2 \eta$ is allowed by the trigonal symmetry. Therefore, if we take for granted that η feels a double-well potential and that a finite ϵ_{IMT}^2 is required to describe the behaviour without monoclinic distortion, we come to the conclusion that, for sufficiently low shear strain ϵ_2 , a local minimum that describes a monoclinic metal unavoidably exists for any value of g and for pure or weakly Cr-doped V_2O_3 . In other words, that conclusion is just consequence of the constraints set by the experiments on the theoretical modelling, and not of an ad hoc choice of the parameters.

Let us now briefly discuss the connection with Tanaka's work, Ref. [47]. Tanaka studied V_2O_3 using dimers as fundamental units. He accurately treated each dimer, and analysed the dimer-dimer coupling in mean-field. In that way he could calculate the values of the tilting angle θ at the local minima of the mean-field energy as function of the dimer length, which he denoted as R , see Fig. 6a of his work that we report here. That figure shows that for any given θ , for instance the red line we have drawn, there are two equilibrium values

Figure 1: Left panel. In orange the V-V dimer, before tilting, the vertical line, and after. However, an additional stretching of the dimer is required for each vanadium to settle in the octahedral void, the blue arrows. The net effect is a vanadium displacement, black arrow, that includes both horizontal and vertical components, the latter being responsible of the larger δ . Both tilting and stretching are here exaggerated for clarity. Right panel. The figure 6a of Tanaka's work, Ref. [47].

of R , compatible with our double-well potential. However, the experimental values of R in the metal, R_{PM} , and insulating, $R_{PI} \simeq R_{AFI}$, phases seem to lie in the same basin of attraction of the potential, as can be inferred from the figure. However, at least for Cr-doped samples, we know that R_{PM} and R_{PI} belong to different wells at $\theta = 0$. Therefore, it is more reasonable that the potential at $\theta \neq 0$ has still one minimum describing the metal and the other the insulator, like, e.g., the blue curve we draw on top of Tanaka's figure. Nonetheless, taking into account the mean-field character of Tanaka's calculation, we do not consider the disagreement a critical issue.

In the revised version, we motivate better the η -functional through the observed behaviour of the structural parameters in Cr-doped samples.

Concerning the other minor points raised by the Referee and related to the theoretical part:

- Grammatical errors or misspellings should be corrected, e.g. "ephasise" [sic], and colloquialisms like "mess up the Mott transition" (line 57) should be avoided. The authors are also

free to select their own title, but the subtext "a richness in disguise" might be substituted for something more concretely descriptive, like "a pathway to metastable metallicity" or similar.

We strove to amend the text and fix all the misspellings. As far as the title is concerned, we modified it accordingly to the referee's suggestion: Nanoscale self-organisation in Mott insulators: a pathway to metastable metallicity.

- Ref 35 and 5 appear to be duplicates

We fixed this issue.

- The number of distinct figures appears excessive and will be problematic for typesetting. Figs. 1-5 might be combined in one figure since they conceptually describe the V_2O_3 structure and transition. I suggest later where other figures may be combined.

We combine Figs. 1, 2 and 3 into a single one, the new Fig. 1. We also combine Figs. 4 and 5 into the single Fig. 2, and added therein the left panel in the figure above, which shows the dimer tilting and the actual displacement of the vanadium endpoints.

- Discussion of x, y, z axes requires these be labeled in Fig. 2

Done.

- Fig. 4: Red/blue for spin orientation should be labeled in a legend, because casual readers comparing Figs. 1-5 with Fig. S1 would be confused that the same colors also indicate V -dimer displacement. Also, the "rotation of dimers" about the b_m axis should be labeled here. Something like the depiction in Fig. 2 of Tanaka would be clearer.

As mentioned above, we added a panel in the new figure 2.

- At line 253, and Eq. (4): The authors rightly decide "not to weigh down the text" with derivations of the Ginzburg-Landau theory, deferring those to the supplement, but since details of the shear strain definition are anyway found there, these might also be removed from the main text. It seems to me the sole message to convey in the main text is that order parameters comprise a 3-fold scalar shear strain and a one-fold scalar dimer distance 'eta' (or 'd', why are two notations used for the same quantity?). If this understanding is correct, then the discussion can be streamlined; if it is incorrect, the authors should clarify.

We followed Reviewer's suggestion. Actually $\epsilon_2 = (\epsilon_{13}, \epsilon_{23})$ behaves as a planar vector, thus a 2-component field. The dimensionless variable η is defined thorough

$$\eta = \frac{1}{d_I - d_M} \left(d - \frac{d_I - d_M}{2} \right), \quad (1)$$

where d_M is the metal equilibrium value, and $d_I > d_M$ the insulator one, and so $\eta = -1/2$ if $d = d_M$, and $\eta = +1/2$ if $d = d_I$. We prefer using η to reduce the number of parameters. We explicitly show the above relation between d and η in the revised version.

- *Line 304 refers the elastic constants which are provided context in the supplement, but not in the main text; here they should be labeled, or better yet their appearance should be deferred entirely to the supplement.*

We indicate explicitly the relation between the functional parameters and the elastic constants in the revised version.

- *Line 238...* According to GGA and GGA+U calculations of Ref. [48], which are supposedly valid at weak coupling, the ambient pressure corundum metal at $T = 0$ is unstable to a monoclinic distortion because of partial Fermi surface nesting, thus implying that c_{44} is negative in the hypothetical zero-temperature corundum metal. The same instability arises if one assumes that V_2O_3 is instead very strongly correlated, and thus GGA not accurate. In that case, one can, e.g., build the effective spin-1 Heisenberg Hamiltonian that describes a Mott insulator of V^{3+} atoms. That turns out to be a highly frustrated $J_1 - J_2$ model on a honeycomb lattice, as also discussed in Ref. [48] and more recently in Ref. [52]. The magnetic ground state of that model breaks C_3 , and therefore is stabilised by the monoclinic distortion. Those results imply that, irrespective of the strength of correlations, theoretical analyses predict that the corundum phase must unavoidably give in to a monoclinic one at low temperature. That is actually the meaning of the obscure sentence "Such weak-coupling Fermi surface instability has its strong coupling counterpart" that we modified in the revised version.

- *The discussion of 'gamma' and 'tau'...* The temperature dependent properties of the electrons are hidden in τ , ϵ_{IMT}^2 and γ that parametrise the effective Born-Oppenheimer potential of the shear strain and dimer length. We do not calculate them explicitly, but just guess their qualitative behaviour from previous calculations and experimental data. As previously discussed, the high temperature corundum phase has an electronic susceptibility towards a monoclinic distortion that must grow lowering T , which affects the effective shear strain stiffness $c_{44} \sim \tau$, driving it towards zero as T goes down.

In turn, since the electron subsystem is known to behave more insulating-like the higher T is, ϵ_{IMT}^2 must diminish upon rising T , eventually crossing zero at the paramagnetic metal-insulator transition in weakly Cr-doped samples. This behaviour was mentioned in the original version, but maybe not enough emphasised.

The temperature dependence of $\gamma \sim c_{14}$ also derives from the electronic instability towards a monoclinic distortion that, in the hexagonal plane, makes one bond longer than the other two. That distortion requires $\gamma < 0$. On the contrary, at high temperature $c_{14} > 0$, which would stabilise a distortion with one bond shorter than the other two; actually observed at high pressure. Therefore, in order to rationalise the ambient pressure behaviour, γ must decrease upon lowering T , and cross zero either before or right at the transition. Once again, such pronounced temperature dependence, observed in Cr-doped samples, is largely contributed by electrons. Also this issue was mentioned in the original version, but maybe

not clearly. We have tried to improve the explanation in the revised version.

We agree that a more reliable approach would be to explicitly calculate the temperature-dependent Born-Oppenheimer potential, as done, e.g., in the two works cited by the Reviewer or in a recent one by one of the co-authors, Phys. Rev. Res. **2**, 013298 (2020). We clearly state that in the revised version. A more reliable approach would not change the shape of the potential, as the Reviewer says, since the shape is just dictated by symmetry, but could allow obtaining more quantitative results.

- *The authors should explain more clearly where Eq. 4 comes from...*

This point has been discussed above. The revised version includes a more detailed motivation of the η -functional.

- *Line 361...* The energies in Fig. 7 actually refer to the depth of the local minima in the energy functional. We explicitly mention that in the revised version, as it was unclear in the original one. Since the energy functional depends on the amplitude ϵ_2 of the shear strain, on the dimensionless variable η related to the dimer length, as well as on τ , we preferred to show the coexistence as in Fig. 7, where one can appreciate the different spinodal points as well as the fact that the monoclinic metal does exist but only as a metastable phase, i.e., a local minimum that never becomes the global one. However, following the Reviewer's suggestion, we include in the revised version the free-energy landscape as a function of η and $\epsilon_2 = |\epsilon_2|$ at $\tau = 10$, where all minima, rhombohedral metal, monoclinic metal and monoclinic insulator, are present.
- *Fig. 8...* The colour plot in Fig. 8 has been done as follows. We recall that the three twins are defined by the strain vectors

$$\epsilon_{2,1} = (+\sqrt{3}/2, +1/2), \quad \epsilon_{2,2} = (-\sqrt{3}/2, +1/2), \quad \epsilon_{2,3} = (0, -1).$$

What we practically do is to build three separate figures. In the first one we plot the y -component of $\epsilon_2(\mathbf{r})$ on a color scale from -1 (plum) to 0 (white). In the second, the x -component from $+\sqrt{3}/2$ (blue) to 0 (white), and in the third still the x -component from $-\sqrt{3}/2$ (orange gold) to 0 (white). Finally, we superimpose the three colour plots. Evidently, when the lighter regions of all three figures overlap that implies both x and y components are equal to zero, thus the undistorted phase. That is a bit intricate, but it was the only way we devised to visualise the two-component strain in a single figure. However, it implies that a color scale palette is cumbersome to show. In the revised version we include the above explanation directly in the caption.

The local strain is calculated self-consistently within mean field, i.e., it is the thermal average of the local pseudo-spin with a local Hamiltonian where the coupling to other sites is absorbed into an effective local pseudo-magnetic field. As such, the outcome is generally not parallel to any of the allowed pseudo-spins, nor has unit length. Put in different words,

if $|n, \mathbf{R}\rangle$, $n = 0, 1, 2, 3$, are the four available states at site \mathbf{R} , the mean-field wavefunction corresponds to $|\psi\rangle = \sum_{\mathbf{R}} \left(\sum_{n=0}^3 \psi_{n\mathbf{R}} |n, \mathbf{R}\rangle \right)$, so that the wavefunction at each site is a superposition of the four available states. In performing the calculations, we use back and forth fast Fourier transform in order to better deal with the long range potential, as mentioned in the Supplementary Material.

- *Can the authors comment...* We believe that the reason is exactly that mentioned by the Reviewer: it is favourable to have a smoother interface at which the amplitude of the strain vanishes rather than a sharp one in which the amplitude remains constant and just the direction suddenly changes. This is evident in Fig. 8.
- *At line 393...* Our mean field scheme gives only indirect access to the space-dependent $\eta(\mathbf{r})$ via the local shear strain amplitude $\epsilon_2(\mathbf{r})^2$, as we explain in the text above line 393 and in the caption of Fig. 9. Consistently, we just show the latter, which we have access to. A fully self-consistent scheme dealing with both $\epsilon_2(\mathbf{r})$ and $\eta(\mathbf{r})$ is unfortunately unfeasible.
- *At line 399...* We thank the Reviewer, and we corrected the mistake.
- *The discussion at line 409...* The one-to-one correspondence between the orbital occupancy and the effective trigonal field has been earlier discussed theoretically by one of the authors in Ref. 29. As the Reviewer states, the influence of the trigonal splitting on the relative occupation of a_{1g} and e_g^π orbitals is clear. However, that correspondence holds also in the reverse direction. One can realise it already at the mean field level. The energy splitting between a_{1g} and e_g^π has a mean field contribution proportional to U times the difference between the average e_g^π occupancy and a_{1g} one, just like the contribution to the spin exchange splitting in a magnet is proportional to the average magnetisation. Therefore, increasing a_{1g} occupancy at the expense of the e_g^π one, as the laser pulse does, reduces the effective trigonal splitting. In the revised version we briefly explain that correspondence.
- *Fig. 9 and 10...* We merged the two figures. The difference between the two reference equilibrium states is just because they are outcome of different calculations. We apologise for that.

2 Time-resolved PEEM experiments – *The time-resolved PEEM results are spectacular and the capability to resolve any real-space features emerging after pump excitation should be given more attention and discussion. The authors seem quick to conclude "the pattern is unchanged," and leave it at that. While this leads to accord with their interpretation, readers like myself are eager to know what new real-space information can be seen or not seen as a consequence of pumping. Origins for transient impacts on the XLD contrast should be given more discussion, and some speculation is welcome. Future experimental directions building on this work should be discussed in the conclusion.*

We are glad that the Reviewer appreciated the novelty of the time-resolved PEEM experiments presented in this work. Indeed, this is the first report of time and spatially resolved experiments that allow to shed light on solid-solid transformations in quantum correlated materials. These results are just the first example of many further experiments that will follow suit in the next years. Nonetheless, the present experimental apparatus has two mainly limitations related to the poor temporal resolution (~ 80 ps), which does not allow to snap the dynamical changes of the monoclinic domains in the early stages of the photoinduced insulator-to-metal transition, and to the poor signal-to-noise-ratio related to the necessity of gating the PEEM signal to synchronize it with the optical excitation. For these reasons we decided to report just the most evident and important result, that is, essentially, the absence of any significant melting of the monoclinic domains of the insulating phase.

Nonetheless, we admit that we were probably a little bit lazy in presenting the time-resolved results and, stimulated by Reviewer's comments, we made an additional and significant effort to analyze the data and infer new spatial information about the metastable state created by the pump excitation. In particular, we added to Fig. 10 of the revised paper the dynamics of the XLD contrast along specific lines perpendicular to the stripe-like monoclinic domains. These data help, on the one hand, to appreciate the small variation of the monoclinic nanotexture induced by the pump excitation and, on the other hand, to link the small change of the XLD contrast to the variation of the a_{1g} occupation expected in the metastable monoclinic phase. In the revised version of the manuscript we added these data as well as an extended additional discussion, based on new calculations reported in the Supplementary Information and on already published cluster multiplet calculations [J.-H. Park et al. Phys. Rev. B 61, 1150611509 (2000)]. Furthermore, we also added in the conclusions a discussion of possible future directions to investigate spatially resolved transitions in correlated materials. We thank again the referee for stimulating us to improve the presentation of these new and interesting data.

Concerning the other minor points raised by the Referee and related to the experimental part:

- *Fig. 6: The position labels could be made more clear with e.g. better contrasting colors. This figure also demands a cartoon of the PEEM setup showing the angle of incidence of X-rays and associated plane where polarization is controllable. A related cartoon appears in a later figure; my forthcoming comment suggest to bring that graphic forward to this figure. Arb. Units on the y-axis of Fig. 6b should be made explicit. What is the overall depth of contrast permissible at the X-ray incidence angle?*

In the revised version we made the cursors larger (see reply to Reviewer 2) to enhance their visibility and to better indicate the area over which the integration of data reported in Fig. 6b is performed. We understand Reviewer's eagerness for further experimental details.

Many of these details have been extensively discussed in our previous work [Ronchi et al, PRB 100, 075111 (2019)] and therefore we wanted to keep the technical description as light as possible. Nonetheless, we understand that for a general reader a more careful introduction of the technique would be useful. Following Reviewer’s suggestion, in the revised version we anticipate the set-up scheme in order to provide the reader with the information to understand the basics of the PEEM experiments (both static and time-resolved). We decided to keep the figures separated to avoid weighing the figure down. As far as the contrast is concerned, the PEEM images are affected by a background that is mainly related to illumination issues. The absolute values of the signals reported in Fig. 6 strongly depends on many parameters and on the background subtraction. In order to appreciate the contrast signal between different domains it is more useful to compare the X-ray Absorption Spectra (XAS), which can be easily normalized. We therefore added the Supplementary Figure S7, which reports the XAS spectra taken on two neighboring monoclinic domains.

- *In Fig. 6a, the inferred pseudo-hexagonal axes should also be labeled for one or more domains, to show how they align with the domain orientations. It is stated that the interface between two of the three monoclinic twins is perpendicular to the bm axis of the third one, can that be visualized here? Would it be less cryptic to state that the habit plane between two twins is actually the complement of their two local y-axes, along which their primary distortions form?*

Following Reviewer’s suggestion we added the pseudo-hexagonal vectors and we rephrased the sentence that describes the formation of the domains.

- *As alluded earlier, the Fig. 11 cartoon should be combined into Fig. 1 to subsume the recommended graphic that would describe the XLD-PEEM setup.*

We followed Reviewer’s suggestion and anticipated the Figure presenting the cartoon of the PEEM setup.

- *In Fig. 13, the authors have claimed the monoclinic nanotexture is fully retained, whereas I can notice some dissimilarities (however slight) or added structure in the comparison of pre- and post-pump images. This is not necessarily to the disservice of the interpretation, but the authors should comment on these differences where they appear. Upon encountering the experimental result, a reader is tempted to find evidence for smearing enhanced foremost at twin boundaries, as predicted by the simulations. From the experimental perspective, is there any possibility to locate attributes of the monoclinic metal in the PEEM images, or is there no straightforward contrast mechanism? Reflections on this question and suggestions for next-generation experiments should be included in the conclusion.*

We thank the Reviewer for having raised these interesting points. It is very difficult to infer the XLD signal of a phase which does not exist at equilibrium. Nonetheless we can trust cluster multiplet calculations [Park et al. PRB 61, 11506 (2000)], which show a dependence

of X-ray absorption on the occupation of the e_g^π and a_{1g} orbitals. This result is confirmed by ad-hoc calculations that we added to Section S4. As explained in the manuscript, to reduce noise and background, each reported image is the difference between images taken with X-ray pulses at 520 eV, for which the contrast between signals from different monoclinic distortions is maximum, and 518 eV, for which the contrast is minimum. If we therefore consider the X-ray absorption difference at these energies, calculations show that the signal is suppressed when the occupation of the initial state changes from e_g^π to e_g^π - a_{1g} . Therefore, the weak suppression of the XLD contrast is compatible with the creation of a transient metastable state with enhanced a_{1g} occupation. In the revised version of the manuscript we added a detailed discussion of this issue.

- *The authors should comment why time-resolved smearing is observable at all- is there any associated feature (or reduction) emerging in the shear strain maps corresponding to Fig. 10? Alternatively, would a monoclinic metal be expected to provide reduced XLD contrast? Lastly, a trivial possibility that should be addressed is where the pump excitation melts the top 20% of the film to the corundum phase, which might be sufficient to yield an entire reflectivity response if the optical penetration depth is equal to or less than (e.g.) 8 nm, and likewise reducing the PEEM contrast. This possibility runs counter to the presented interpretation. The thin film provides a best platform to avoid such effects, but the authors should suitably dismiss such possibilities with analysis in reference to relevant V2O3 optical constants. For comparison, the relevant penetration depth of the soft X-rays (or escape depth of photoelectrons) should also be indicated.*

As argued just above, the smearing of the signal is connected to the change of the occupation of the e_g^π and a_{1g} orbitals.

As far as the homogeneity of the excitation is concerned, the pump penetration depth is of the order of 300 nm if we consider the optical properties taken from the literature [Qazilbash et al PRB 77, 115121 (2008)]. The pump excitation can thus be safely considered as homogeneous within the 50 nm film thickness. On the other hand, the surface sensitivity of the PEEM technique in the total electron yield configuration is difficult to evaluate because it depends on the material and on the X-ray energy. An escape depth of 3-5 nm is reported in the literature [T.J. Regan et al. PHYSICAL REVIEW B, VOLUME 64, 214422 (2001); S. Gota et al. PHYSICAL REVIEW B 62 4187 (2000)] for similar oxides and can be considered as a reasonable approximation. We added these considerations to the discussion.

- *The comparison with V2O3 on Cr2O3 is an impressive achievement, particularly since the resultant nanotextures are quite comparable to the pure V2O3 film. The authors should clarify the putative relationship between ‘ ϵ_{IMT} ’ and the lattice constant ‘ a ’, and Fig. 14b should include a vertical trace for the native V2O3 lattice constant to which strained values should be compared. Were the time-resolved PEEM results similar for V2O3 on Cr2O3?*

We thank the Reviewer for appreciating the comparison between different strained films. At this level, the relation between residual strain and the value of ϵ_{IMT} , which controls the monoclinic metallic phase, is empirical and it was originally led by the idea that if the a_H axis is already stretched as compared to native V2O3, this could favour the emergence of metastable metallicity with monoclinic distortion. These preliminary results clearly call for the development of microscopic models that can directly relate ϵ_{IMT} to microscopic quantities, such as the lattice constants and the residual strain. In the revised version of the manuscript we included these considerations in the final discussion and conclusions, as suggested by the referee.

As far as the lattice constant of native V2O3, i.e. $a_H=4.936 \text{ \AA}$, it is reported in the main text (Sec. 2). We avoid to report it in Fig. 11b of the revised version (Fig. 14 b of the previous version) because it would force us to change the scale in a way that would make the difference between the different investigated samples less evident. Finally, the beamtime dedicated to time-resolved experiments was so limited and the experiment so demanding that we had the possibility to measure just one sample. We hope to repeat soon similar experiments on a larger set of samples.

- *Line 503 asserts, the domain boundaries stabilize and protect the photoinduced non-thermal monoclinic metallic state, which would be unstable at equilibrium an in [sic] homogeneous systems. This seems to be one of the important implications of a c-cut film which enables the twin texture. We might suppose films with other growth directions would have fewer (or no) twin domains. Can the authors point to evidence the photo-induced response of the c-cut film is therefore qualitatively different (e.g. lower pumping threshold) from films at other growth planes, which likely have fewer twins? The authors should comment on the possibility to engineer twin domains as a more general recipe for encouraging intermediate states - could the twin texture be made even more fine-grained and if so by what means? In what other systems besides V2O3 might twin-stabilized intermediary states be possible?*

We thank the Reviewer for the suggestions that indeed are well within our future research plans. Comparing the threshold of time-resolved experiments in different samples is always tricky because it requires an accurate control of the experimental conditions and of the domains topology. The complete set of data we have accumulated so far is just on the two samples described in the manuscript, which have been grown via MBE in the same conditions and fully characterized by X-ray diffraction and electrical measurements. The comparison we show is intended to demonstrate that, given the same nanotexture, the residual strain can be used to control ϵ_{IMT} and therefore the photoinduced metastable state. On the other hand, we agree that crystals cut along different planes should exhibit less or no nanotexture at all, which would modify the photoinduced metallicity. However, a clear response on that would require an extensive campaign, including PEEM, time-resolved, electrical and X-ray diffraction measurements, that goes beyond the present

effort. This is definitely one of the directions opened by our work. In the revised version, as suggested by the Reviewer, we included this discussion as well as an overview of other systems that could exhibit nanotexture-stabilized metastable phases.

We hope that the revised version and the discussion above satisfy the Reviewer, whom we thanks again for her/his very helpful comments.

REVIEWERS' COMMENTS

Reviewer #1 (Remarks to the Author):

I am satisfied with the Authors' response and can now recommend publication of the manuscript in Nature Communications.

Reviewer #2 (Remarks to the Author):

In their rebuttal letter and revised manuscript, the authors have addressed all my questions and comments satisfactorily, and improved the presentation of the paper considerably. I can now recommend this work for publication in Nature Communications.